# Acylation of glycerolipids in mycobacteria

Shiva Kumar Angala[1], Ana Carreras-Gonzalez[2,3], Emilie Huc-Claustre[1], Itxaso Anso [4], Devinder Kaur[1,8], Victoria Jones[1], Zuzana Palčeková[1], Juan M. Belardinelli[1], Célia de Sousa-d'Auria[5], Libin Shi[1], Nawel Slama[6], Christine Houssin [5], Annaïk Quémard [6], Michael McNeil[1], Marcelo E. Guerin [2,3,4,7,9] & Mary Jackson [1] ✉

We report on the existence of two phosphatidic acid biosynthetic pathways in mycobacteria, a classical one wherein the acylation of the *sn*-1 position of glycerol-3-phosphate (G3P) precedes that of *sn*-2 and another wherein acylations proceed in the reverse order. Two unique acyltransferases, PlsM and PlsB2, participate in both pathways and hold the key to the unusual positional distribution of acyl chains typifying mycobacterial glycerolipids wherein unsaturated substituents principally esterify position *sn*-1 and palmitoyl principally occupies position *sn*-2. While PlsM selectively transfers a palmitoyl chain to the *sn*-2 position of G3P and *sn*-1-lysophosphatidic acid (LPA), PlsB2 preferentially transfers a stearoyl or oleoyl chain to the *sn*-1 position of G3P and an oleyl chain to *sn*-2-LPA. PlsM is the first example of an *sn*-2 G3P acyltransferase outside the plant kingdom and PlsB2 the first example of a 2-acyl-G3P acyltransferase. Both enzymes are unique in their ability to catalyze acyl transfer to both G3P and LPA.

Phospholipids populate both the inner and outer membranes of mycobacteria[1,2]. While naturally occurring bacterial phospholipids generally harbor C16:0 or C18:1 fatty acyl chains at position 1 and unsaturated fatty acids (C16:1 or C18:1) at position 2 of the glycerol moiety, mycobacterial phospholipids are unusual in having unsaturated or branched oleyl (C18:1) or tuberculostearoyl (C19) substituents principally esterifying position 1, and palmitoyl (C16:0) principally occupying position 2[3]. The same fatty acid distribution is found in mycobacterial triglycerides (TAG)[2,4]. How this unusual positional distribution of acyl chains influences the physiology of mycobacteria, including the fluidity, permeability and various other functions of their cell envelope remains unclear[5]. Moreover, the biosynthetic origin of this compositional oddity is not known since none of the enzymes

involved in the acylation of the *sn*-1 and *sn*-2 positions of mycobacterial glycerolipids have yet been formally characterized despite the finding of matching activities in mycobacterial cell-free extracts and evidence that expression of the glycerol-3-phosphate (G3P) acyltransferase candidate PlsB1 (Rv1551) from *Mycobacterium tuberculosis* (*Mtb*) in *Escherichia coli* stimulates the cell-free synthesis of phospholipids[6–8].

Bacterial glycerophospholipids and TAG are synthesized in the cytoplasmic membrane from the common precursor phosphatidic acid (PA)[9–11]. G3P, which is formed by reduction of the glycolytic pathway intermediate dihydroxyacetone phosphate, is successively acylated at the *sn*-1 and *sn*-2 positions, in reactions catalyzed by G3P acyltransferase and 1-acyl-G3P acyltransferase (or lysophosphatidic acid [LPA] acyltransferase), respectively, yielding PA [Fig. 1a]. Two

[1]Mycobacteria Research Laboratories, Department of Microbiology, Immunology and Pathology, Colorado State University, Fort Collins, CO 80523-1682, USA. [2]Unidad de Biofisica, Centro Mixto Consejo Superior de Investigaciones Cientificas - Universidad del País Vasco/Euskal Herriko Unibertsitatea (CSIC-UPV/EHU), Barrio Sarriena s/n, Leioa, Bizkaia 48940, Spain. [3]Departamento de Bioquímica, Universidad del País Vasco, Leioa, Spain. [4]Structural Glycobiology Laboratory, Biocruces Bizkaia Health Research Institute, Cruces University Hospital, Barakaldo, Bizkaia 48903, Spain. [5]Université Paris-Saclay, CEA, CNRS, Institute for Integrative Biology of the Cell (I2BC), Gif-sur-Yvette, France. [6]Institut de Pharmacologie et de Biologie Structurale (IPBS), CNRS, UPS, Université Toulouse III - Paul Sabatier, Toulouse, France. [7]IKERBASQUE, Basque Foundation for Science, 48009 Bilbao, Spain. [8]Present address: New England Newborn Screening Program, University of Massachusetts Medical School, Worcester, MA 01605, USA. [9]Present address: Structural Glycobiology Laboratory, Department of Structural and Molecular Biology, Molecular Biology Institute of Barcelona (IBMB), Spanish National Research Council (CSIC), Barcelona Science Park, c/Baldiri Reixac 4-8, Tower R, 08028 Barcelona, Catalonia, Spain. ✉e-mail: Mary.Jackson@colostate.edu

distinct families of acyltransferases are responsible for the acylation of position sn-1 of G3P in bacteria. PlsB-type acyltransferases, primarily use the acyl-ACP products of Fatty Acid Synthase II (FAS-II) as acyl donors although PlsB enzymes from γ-proteobacteria may also use acyl-CoAs from exogenous fatty acids. The more widespread PlsY-type acyltransferases, in contrast, utilize acyl-phosphates produced from acyl-ACPs by PlsX enzymes as acyl donors[9–12]. The acylation of position sn-2 of LPA is carried out by PlsC enzymes that use acyl-ACPs as acyl donors although PlsCs from γ-proteobacteria may also use acyl-CoAs.

Evidence based on the analysis of mycobacterial genomes and crude enzyme assays suggests that mycobacteria may diverge from this universal biosynthetic scheme in several respects. Firstly, while BLAST searches for plsB, plsC, plsX and plsY genes in the sequenced genomes of Mtb H37Rv and Mycobacterium smegmatis (Msmg) mc²155 identified candidate genes for the plsB and plsC families of acyltransferases, they failed to reveal any plsX and plsY homologs suggesting that fast- and slow-growing mycobacteria exclusively rely on the PlsB/PlsC pathway for phospholipid and TAG synthesis. To this date, archaea, eukarya and Xanthomanadales in γ-proteobacteria are the only described organisms known to not contain plsX and plsY homologs[11]. Secondly, owing to the coexistence of a fatty acid synthase (FAS) I and a FAS-II system in mycobacteria, the former being responsible for the synthesis of C16 to C26 acyl-CoAs and the latter being responsible for the elongation of longer acyl-ACP substrates used in the biosynthesis of mycolic acids[4,13], mycobacteria are thought to use acyl-CoAs generated by FAS-I rather than acyl-ACPs originating from FAS-II in the synthesis of their phospholipids and TAG[2,7]. Thirdly, pioneering studies by Okuyama et al[7]. indicated that the conversion of G3P to PA in Mycobacterium butyricum membranes may proceed through two simultaneously operating acylating pathways wherein the first acylation may take place either at position sn-1 or at position sn-2 of the glycerol moiety. The first acylation occurring at position 2 of G3P was found to be favored under the conditions of this study and to preferentially incorporate C16:0 yielding 2-acyl-G3P. This product was subsequently acylated at position sn-1 with a clear preference for oleic acid. According to this study, this atypical pathway wherein the acylation of position sn-2 of G3P precedes that of position sn-1 thus seemed to hold the key to the distinctive acylation pattern of PA and derived glycerolipids produced by mycobacteria. Here, we report on the identification of the acyltransferases responsible for this unique pathway to glycerolipid synthesis.

## Results

### Candidate glycerolipid acyltransferases in *Mtb* and *Msmg*

A BLAST search for candidate glycerolipid acyltransferases in the sequenced genome of Mtb failed to reveal any PlsY or PlsX homologs suggesting that, contrary to the situation in most prokaryotes, glycerolipid biosynthesis in this species does not involve acyl-phosphate substrates. The search, however, revealed the existence of several as yet uncharacterized proteins displaying the characteristics of PlsB (Enzyme Commission [EC] 2.3.1.15; gene ontology GO 0004366) and PlsC (EC 2.3.1.51; gene ontology GO 0003841) enzymes [Table S1]. Neither of the two PlsB candidates, PlsB1 and PlsB2, are predicted to be required for Mtb growth by saturation transposon mutagenesis[14] suggesting that they share at least partially redundant activities. While cell-free assays in E. coli support the involvement of PlsB1 in phospholipid synthesis[6], only PlsB2 (Rv2482c) appears to be conserved in mycobacteria [Table S1]. Of the seven PlsC homologs, four are conserved in Msmg (Rv0502, Rv2182c, Rv2483c and Rv3816c) and in Mycobacterium leprae, a species considered to have a minimal mycobacterial genome. Of these, only Rv2182c is predicted to be required for growth[14]. Rv2182c harbors the highly conserved active site motif HX₄D of glycerolipid acyltransferases (H41-X₄-D46) and maps, in all mycobacterial genomes analyzed to date, adjacent to genes involved in the biosynthesis of phosphatidylinositol mannosides (PIM), lipomannan and lipoarabinomannan (pimB' [Rv2188c], mptA [Rv2174] and mptC [Rv2181c])[15–18] [Table S1]. It is not predicted to contain any transmembrane domains[19]. Rv2182c shares 75% amino acid identity (83% similarity) with its ortholog in Msmg (MSMEG_4248). We renamed this enzyme PlsM for Pls enzyme from Mycobacteria.

### Disruption of *MSMEG_4248* in *Msmg* abolishes growth

The question of the essentiality of the putative mycobacterial PlsC enzyme, MSMEG_4248 (PlsMsmg), was approached genetically, that is gene inactivation in Msmg was attempted in the presence or absence of a rescue copy of this gene. Gene knock-out (KO) at the plsMsmg locus of Msmg was only achievable in the presence of a wild-type (WT) copy of plsMsmg or plsMtb (Rv2182c) expressed from replicative plasmids [Fig. 1b]. plsMsmg and plsMtb thus display analogous functions and appear to be essential for the in vitro growth of Msmg and possibly Mtb[14].

### Effect of silencing *plsM* on the glycerolipid content of *Msmg*

Construction of conditional mutants of Msmg in which a rescue copy of plsMsmg or plsMtb was placed under control of an anhydro-tetracycline (ATc)-inducible promoter revealed that the merodiploid strains rapidly ceased growing when placed on agar or liquid medium devoid of ATc where the expression of the rescue copy was lost [Fig. 1c, d]. Importantly, gene silencing led to a dramatic decrease in the phospholipid content of the conditional mutants [Fig. 1e]. The LC/MS-based analysis of individual glycerolipid species in the conditional knock-down MsmgΔplsM/pSETetR-plsMsmg and control strain (WT Msmg harboring an empty pSETetR plasmid) grown to the same OD₆₀₀ nm in 7H9-OADC-tyloxapol in the presence of 50 ng/mL or 1 ng/mL ATc is shown in Fig. 1f. plsM silencing in MsmgΔplsM/pSETetR-plsMsmg led to significant decreases in diacylglycerides (DAG), phosphatidylethanolamine (PE), lyso-PE, phosphatidylinositol (PI) and cardiolipin (CL) that were not observed in the control strain. TAG was also significantly reduced in the conditional knock-down relative to the control strain at both ATc concentrations with the difference between the two strains becoming more pronounced at the lower ATc concentration. Remarkably, these quantitative changes in phospholipid content were accompanied by an increase in unsaturated acyl chains esterifying all forms of glycerolipids in the conditional mutant grown at the low ATc concentration when compared to MsmgΔplsM/pSETetR-plsMsmg grown in the presence of 50 ng/mL ATc or to the control strain at both ATc concentrations [Fig. 1g]. Consistent with these changes, GC/MS analysis of total fatty acid methyl esters in the conditional mutant revealed a 32% increase in the relative abundance of oleic acid in the cells that accompanied a 58% decrease in tuberculostearic acid (C19), a 21% decrease in palmitic acid and an 87% increase in the relative abundance of longer, C24:0, fatty acyl chains [Table S2].

Altogether, the data are thus consistent with plsM playing an essential role in the early stages of glycerolipid synthesis.

### Complementation with *plsC* from *E. coli* restores *MsmgΔplsM* growth

Given the homology of PlsM with bacterial PlsCs, we first resorted to a cross-complementation approach to determine whether well-characterized LPA acyltransferases such as the PlsC enzymes from E. coli and B. subtilis could rescue the growth of MsmgΔplsM. PlsC from E. coli may use both acyl-CoAs and acyl-ACPs substrates in the acylation of the sn-2 position of LPA[20,21] whereas the B. subtilis PlsC enzyme relies exclusively on acyl-ACP donors[22]. Remarkably, disruption of the plsM locus of Msmg was achievable in the presence of the plsC gene from E. coli (plsCcoli) [Fig. 2a] but not that from B. subtilis (plsCsubtilis) despite comparable levels of expression of plsCcoli and plsCsubtilis in Msmg [Fig. S1]. Consistent with the ability of plsCcoli to rescue the mutant, de novo phospholipid biosynthesis was restored in MsmgΔplsM/pMVGH1-

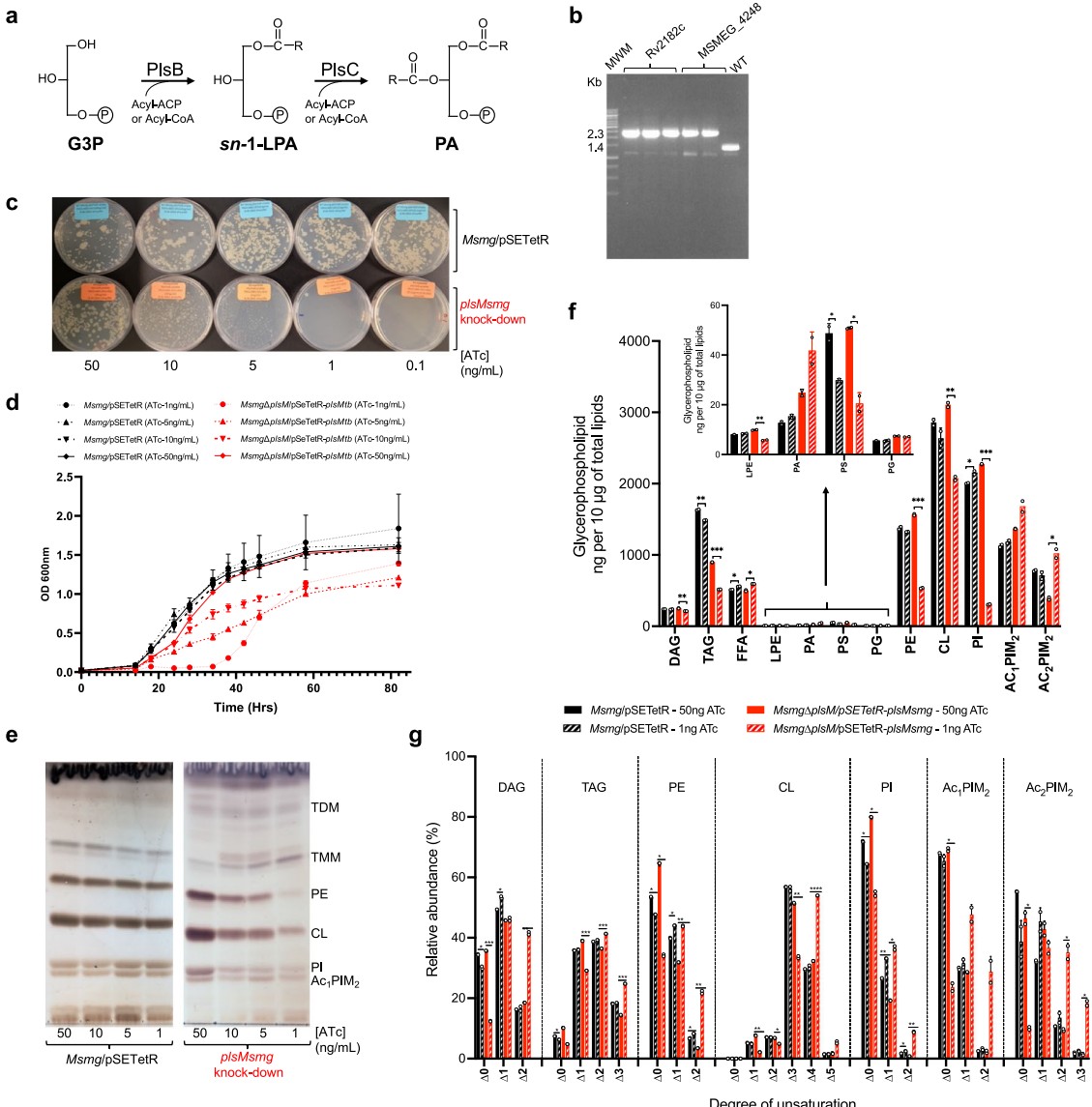

**Fig. 1 | *plsM* silencing in *M. smegmatis* abolishes phospholipid synthesis and leads to growth arrest. a** The PlsB/PlsC pathway to PA formation in *E. coli*. The PlsB-catalyzed transfer of a fatty acid to position *sn*-1 of G3P from acyl-ACP or acyl-CoA precedes the acylation of position *sn*-2 of *sn*-1-lysoPA by PlsC. **b** Allelic replacement at the *plsM* locus of *Msmg* mutants rescued with the *plsM* orthologs of *Msmg* (*MSMEG_4248*) and *Mtb* (*Rv2182c*) expressed from replicative pSETetR plasmids under control of an ATc-inducible (TET-ON) promoter was confirmed by PCR in two to three independent clones. The WT 1,394-bp amplification signal is replaced by a 2,312-bp fragment in the mutants due to the insertion of a 1.2 kb-kanamycin resistance cassette between the PstI and NruI restriction sites of *MSMEG_4248*. **c** Growth of the *MsmgΔplsM*/pSETetR-*plsMtb* conditional knock-down (red symbols) and *Msmg*/pSETetR control strain (black symbols) on 7H11-OADC plates and in, **d**, 7H9-ADC-tyloxapol at 37 °C in the presence of different concentrations of ATc. Shown in (d) are the means +/- SD of triplicate cultures (*n* = 3 biologically independent samples). Growth is totally inhibited in liquid culture in the conditional mutant at 1 ng/mL ATc until ATc regulation is lost (~40 h post-inoculation) and the strain starts replicating at a comparable rate to the control strain. **e** The phospholipid content of *Msmg* control and *MsmgΔplsM*/pSETetR-*plsMtb* cells grown on 7H11-OADC agar plates as shown in (**c**) were analyzed by TLC in the solvent system CHCl₃:CH₃OH:H₂O (65:25:4 by vol.). The cells were collected on the same day and ~50 μg of total lipids were loaded per lane. **f** Lipids from *Msmg* control and *MsmgΔplsM*/pSETetR-*plsMsmg* duplicate cultures grown under permissive (50 ng/mL ATc) and non-permissive (1 ng/mL ATc) conditions in 7H9-OADC-tyloxapol at 37 °C to the same $OD_{600}$ (~0.5–0.6) were quantitatively analyzed by LC/MS as described under Methods and the abundance of glycerolipids in the two strains under both culture conditions is shown as means ± SD of *n* = 2 biologically independent samples. **g** Relative abundance (in percentages) of saturated and unsaturated species within each glycerolipid category (DAG, TAG, PE, CL, PI, Ac₁PIM₂ and Ac₂PIM₂) in the same *Msmg*/pSETetR and *MsmgΔplsM*/pSETetR-*plsMsmg* strains grown under permissive (50 ng/mL ATc) and non-permissive (1 ng/mL ATc) conditions as in (**f**). Results are shown as means ± SD of *n* = 2 biologically independent samples. In panels (**f**, **g**) asterisks denote statistically significant differences between culture conditions pursuant to the two-sided unpaired Student's *t*-test (**f** *p < 0.05, **p < 0.005, ***p < 0.0005; **g** *p < 0.01, **p < 0.001, ***p < 0.0001). The results presented in (**c–g**) are representative of two to three independent experiments. CL cardiolipin, DAG diglycerides, FFA free fatty acids, G3P glycerol-3-phosphate, *sn*-1-LPA *sn*-1-lysophosphatidic acid, LPE lysophosphatidylethanolamine, PA phosphatidic acid, PE phosphatidylethanolamine, PG phosphatidylglycerol, PI phosphatidyl-*myo*-inositol, PS phosphatidylserine, Ac₁PIM₂ triacylated forms of phosphatidyl-*myo*-inositol dimannosides, Ac₂PIM₂ tetraacylated forms of phosphatidyl-*myo*-inositol dimannosides, TMM trehalose monomycolates, TDM trehalose dimycolates. Source data for panels (**d**, **f**, **g**) are provided as a Source Data file.

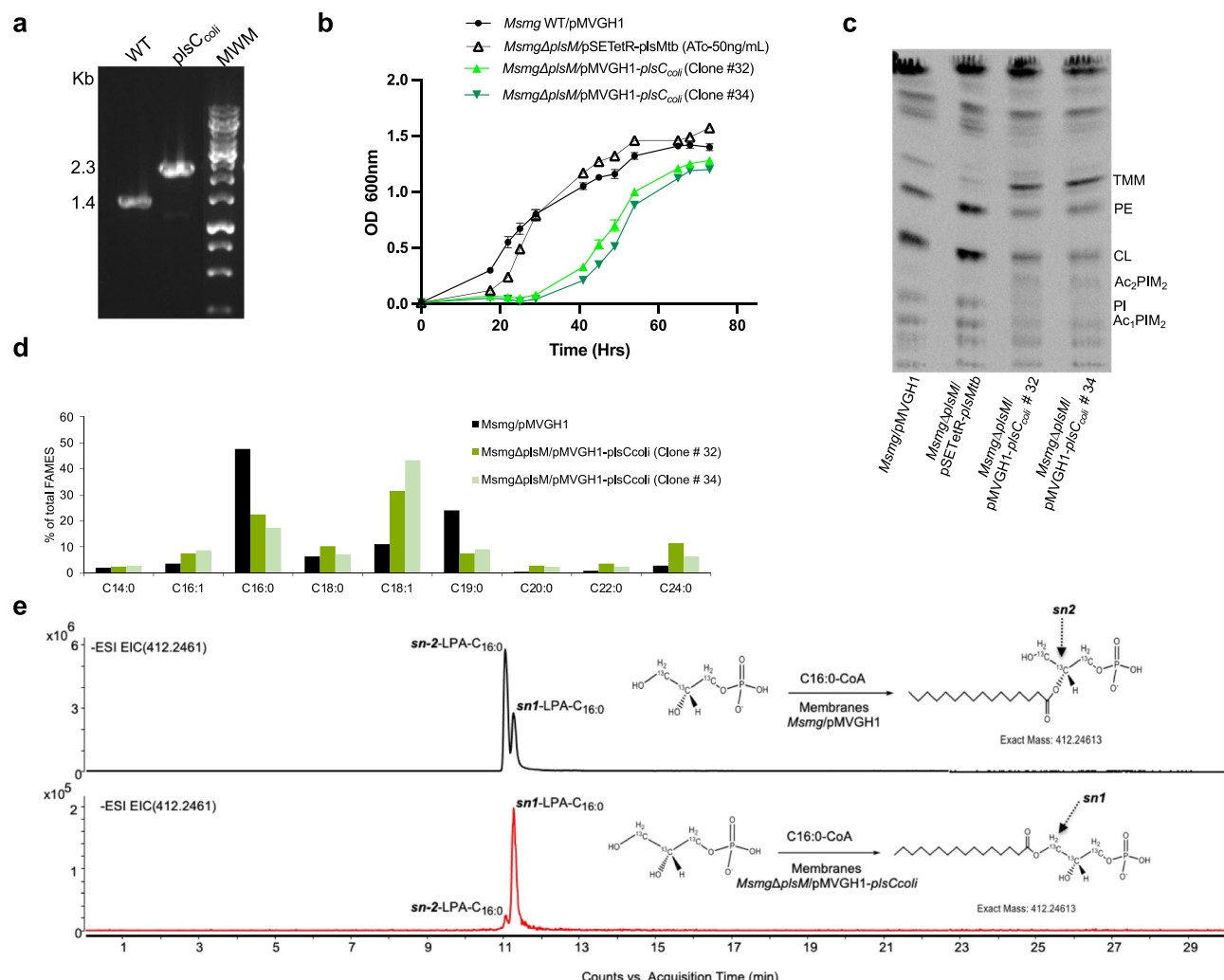

**Fig. 2 | Effect of replacing PlsM by PlsC from *E. coli* in *Msmg*. a** Allelic replacement at the *plsM* locus of *Msmg* mutants rescued with *plsCcoli* was analyzed as in Fig. 1b. **b** Growth characteristics of WT *Msmg* mc²155 (black circles), *MsmgΔplsM*/pSETetR-*plsMtb* (black triangles), and *MsmgΔplsM*/pMVGH1-*plsCcoli* (clones # 32 and 34; light and dark green triangles, respectively) in 7H9-ADC-0.05% tyloxapol at 37 °C (in the presence of 50 ng/mL ATc for *MsmgΔplsM*/pSETetR-*plsMtb*). Shown are the means ± SD of three independent growth curves (*n* = 3 independent biological triplicate) for each strain. **c** De novo phospholipid synthesis in WT *Msmg* mc²155, *MsmgΔplsM*/pMVGH1-*plsCcoli* (clones # 32 and 34) and *MsmgΔplsM*/pSETetR-*plsMtb* grown in 7H9-ADC-0.05% tyloxapol at 37 °C (in the presence of 50 ng/mL ATc for *MsmgΔplsM*/pSETetR-*plsMtb*). [1,2-¹⁴C]acetic acid was added to bacterial cultures when they reached an OD₆₀₀ nm of 0.8 after which cultures were incubated for another 8 h at 37 °C with shaking. [1,2-¹⁴C]acetic acid-derived lipids were analyzed by TLC in the solvent system CHCl₃:CH₃OH:H₂O (70:20:2 by vol.). The same total

counts (dpm) were loaded per lane. Ac₂PIM₂ are tetraacylated forms of phosphatidyl-*myo*-inositol dimannosides. Other lipid abbreviations are as in Fig. 1. The results shown are representative of two independent experiments. **d** Fatty acid composition of WT *Msmg* mc²155 and *MsmgΔplsM*/pMVGH1-*plsCcoli* (two different clones, # 32 and # 34; dark and light green bars, respectively) grown in 7H9-ADC-0.05% tyloxapol at 37 °C (same medium as used in panels b and c). C19:0: tuberculostearic acid. **e** Extracted ion chromatograms (EICs) showing the lysophosphatidic acid (LPA) products resulting from the incubation of membranes prepared from *Msmg*/pMVGH1 and *MsmgΔplsM*/pMVGH1-*plsCcoli* (clone # 32) with [¹³C]-G3P and C16:0-CoA. *sn*-2 palmitoyl transferase activity is dominant in *Msmg*/pMVGH1 whereas it is barely detectable in *MsmgΔplsM*/pMVGH1-*plsCcoli*. C16:0 in the latter strain is essentially transferred to position *sn*-1 of G3P. Source data for panels (**b** and **d**) are provided as a Source Data file.

*plsCcoli* albeit at a lower rate than observed in WT *Msmg* mc²155 harboring an empty pMVGH1 plasmid and *MsmgΔplsM*/pSETetR-*plsMtb* grown in the presence of 50 ng/mL ATc, most likely accounting for the delayed growth (i.e., longer lag period) of the *plsCcoli* rescued mutant [Fig. 2b, c]. The steady-state glycerolipid content of the WT strain and *plsCcoli*-rescued mutant were otherwise similar, except for a slight (~20%) relative increase in TAG content and ~25% relative decrease in CL content in the latter strain [Fig. S2A] and the fact that the *plsCcoli*-rescue mutant presented glycerolipids with relatively more unsaturated fatty acyl chains compared to WT *Msmg* [Fig. S2B] as was observed upon silencing of *plsM* in *Msmg* [Fig. 1g]. Changes in the glycerolipid acylation pattern of *plsCcoli*-expressing *Msmg* mutants reflected in their total fatty acid composition in that, relative to WT

*Msmg* mc²155, a dramatic decrease in C16:0 and C19 concomitant with an increase in C18:1 was noticeable in *MsmgΔplsM*/pMVGH1-*plsCcoli* cells [Fig. 2d]. *plsCcoli*-complemented mutants also tended to accumulate longer (C20:0-C24:0) fatty acyl chains [Fig. 2d].

Despite quantitatively comparable total lipid contents in the *plsCcoli* complemented mutant and the parent WT *Msmg* strain (extractable lipids represent 10.2 ± 0.6 % of the dry cell weight in WT *Msmg*, 9.4 ± 0.4 % in *MsmgΔplsM*/pMVGH1-*plsCcoli*; average ± standard deviation of three independent log phase cultures), the complemented strain showed reduced sliding motility on M63-Tween 80 agar [Fig. S3A] and reduced biofilm-forming capacity in Sauton's medium [Fig. S3B] without presenting any noticeable changes its drug susceptibility pattern [Table S3].

The ability of *plsC*$_{coli}$ but not *plsC*$_{subtilis}$ to partially rescue the growth of *Msmg*Δ*plsM* suggests that acyl-CoAs, rather than acyl-ACPs, are being used as donor substrates in the acylation of LPA in mycobacteria, consistent with FAS-I generating C16/C18-CoAs. The reduced rate of glycerolipid synthesis in the complemented *Msmg* mutant may reflect the insufficient level of expression of *plsC*$_{coli}$ and/or differences in the substrate specificity of the mycobacterial and *E. coli* enzymes. Indeed, in light of the different positional distribution of acyl chains in mycobacterial and *E. coli* glycerolipids whereby position 2 of the glycerol moiety is predominantly occupied by palmitic acid in mycobacteria and by palmitoleic or oleic acid in *E. coli*[11], the marked decrease in C16:0 and relative increase in C18:1 noted in *Msmg*Δ*plsM*/pMVGH1-*plsC*$_{coli}$ cells compared to WT *Msmg* likely reflects the preference of PlsC$_{coli}$ for unsaturated acyl donors. A correlate of this is that *plsM* most likely encodes the acyltransferase responsible for the acylation of position-2 of glycerolipids with palmitic acid. This assumption is further supported by the relative decrease in saturated forms of glycerolipids produced by *Msmg* when the expression of *plsM* is silenced [Fig. 1g].

## Substrate specificity of purified PlsM

The substrate specificity of PlsM was investigated through in vitro assays using purified recombinant PlsMsmg produced in *E. coli* [Fig. S4]. Attempts to similarly produce the PlsM enzyme from *Mtb* essentially yielded insoluble protein and were not pursued further. Enzyme assays first aimed at comparing the efficacy of C16:0-CoA, C18:0-CoA and C18:1-CoA as acyl donors and that of various 1-acyl-2-hydroxy-*sn*-G3Ps and G3P as acceptor substrates. Unexpectedly, the LC/MS analysis of enzyme products revealed that PlsMsmg could efficiently use both types of acceptor substrates, albeit with an apparent preference for 1-acyl-2-hydroxy-*sn*-G3Ps [Fig. 3; Table S4]. Whatever the acyl-CoA donor, 1-palmitoyl-2-hydroxy-*sn*-G3P and 1-stearoyl-2-hydroxy-*sn*-G3P were preferred LPA acceptors over 1-oleoyl-2-hydroxy-*sn*-G3P. C16:0-CoA was the preferred acyl donor whatever the acceptor substrate and this preference was especially marked when 1-oleoyl-2-hydroxy-*sn*-G3P or G3P served as acceptors. In light of the fact PlsM presents the characteristics of a 1-acyl-G3P acyltransferase (PlsC-type enzyme) and, thus, of an enzyme transferring acyl chains specifically to position *sn*-2 [Table S1], the formation of PA (i.e., a product acylated at both positions *sn*-1 and *sn*-2) in reactions that used G3P as the acceptor substrate and C16:0-CoA as the acyl donor was unexpected [Fig. 3a, b]. We tentatively attribute this result to the non-enzymatic intramolecular migration of the acyl chain at the *sn*-2 position to the *sn*-1-position[23,24] despite precautions taken in our enzyme assays and sample preparation to minimize this phenomenon (see Methods). This assumption was supported by the finding of both 1-hydroxy-2-acyl-*sn*-G3P (dominant product) and 1-acyl-2-hydroxy-*sn*-G3P in the products of PlsMsmg reactions that used G3P as an acceptor and C16:0-CoA as the acyl donor [Fig. 3c(a–c)]. Given the ability of PlsMsmg to use 1-acyl−2-hydroxy-*sn*-G3Ps as acceptor substrates [Fig. 3b], the product resulting from the spontaneous transmigration of C16:0 from position *sn*-2 to position *sn*-1 of LPA likely became an acceptor for the PlsMsmg-mediated transfer of a second C16:0 acyl chain onto position *sn*-2, yielding PA.

Radiolabeled assays using [14C(U)]G3P and C16:0-CoA confirmed G3P as a potent acceptor substrate in the palmitoylation reaction catalyzed by PlsMsmg [Fig. S5A] and demonstrated the specificity of this enzyme for C16:0-CoA over C16:0-ACP [Fig. S5B]. Palmitoyl transfer onto G3P was both enzyme- [Fig. S5C] and C16:0-CoA-concentration dependent [Fig. S5B].

Collectively, our results establish PlsMsmg as a unique acyltransferase capable of transferring acyl chains from their corresponding acyl-CoAs, with a preference for C16:0-CoA, onto position *sn*-2 of both G3P and 1-acyl-G3P. The rapid transmigration of acyl chains from position *sn*-2 to position *sn*-1 of LPA makes it impossible to exclude that PlsM may transfer acyl chains to position *sn*-1 in addition to position *sn*-2 of G3P. A dual positional specificity of the enzyme, however, seems unlikely based on the results presented in Fig. 4 that show that when 1-hydroxy-2-palmitoyl-*sn*-G3P is used as an acceptor substrate, PlsMsmg is apparently unable to transfer C18:0 or C18:1 from their respective acyl-CoA donors to position *sn*-1.

Of note, whereas *sn*-2 palmitoyl transferase activity on G3P was detectable in membranes prepared from *Msmg* harboring an empty pMVGH1 plasmid as reported previously[7], membrane preparations from *Msmg*Δ*plsM*/pMVGH1-*plsC*$_{coli}$ were essentially devoid of such activity and, instead, transferred C16:0 to position *sn*-1 of G3P [Fig. 2e]. This result indicates that PlsM is most likely the sole acyltransferase capable of transferring C16:0 onto position *sn*-2 of G3P in *Msmg*. Consistent with the marked preference of PlsM for C16:0-CoA as the acyl donor [Fig. 3b], no 1-hydroxy-2-oleoyl-*sn*-G3P was detectable in membrane prepared from *Msmg*/pMVGH1 (or *Msmg*Δ*plsM*/pMVGH1-*plsC*$_{coli}$) when C18:1-CoA was used as the acyl donor [Fig. S6].

## PlsB2-mediated acyl transfer to the *sn*-1 position of G3P and LPA

Efforts to identify the dominant PlsB-type enzyme responsible for the transfer of acyl chains to position *sn*-1 of G3P next turned to PlsB2 (Rv2482c), one of the two PlsB homologs encoded by the *Mtb* genome and the only one conserved across *Mycobacterium* species, including *M. leprae* [Table S1]. PlsB2 from *Msmg* (PlsB2smg) was successfully produced and purified from *E. coli* [Fig. S4]. Assays using G3P as the acceptor substrate indicated that PlsB2smg efficiently transfers C18:0 and C18:1 and, to a much lesser extent C16:0, from their respective acyl-CoAs to position *sn*-1 of G3P [Fig. 3a–c; Table S4]. In contrast to PlsMsmg, PlsB2smg did not display significant 1-acyl-G3P acyltransferase activity on any LPA acceptor when C16:0-CoA and C18:0-CoA were used as the acyl donors [Fig. 3b]. However, PlsB2smg clearly demonstrated the ability to transfer C18:1 and, to a much lesser extent, C18:0 [Fig. 3b] from their respective acyl-CoA donors to 1-hydroxy-2-palmitoyl-*sn*-G3P generating PA with C18:1 or C18:0 chains in the *sn*-1 position [Fig. 5]. The positional specificity and apparent preference of PlsB2 for C18:0-CoA and C18:1-CoA over C16:0-CoA suggest that PlsB2 is the G3P- and 2-acyl-G3P acyltransferase responsible, together with PlsM, for the formation of glycerolipids harboring palmitic acid in position *sn*-2 and stearic or oleic acid (or tuberculostearic acid) at position *sn*-1. Tuberculostearic acid is known to arise from the *S*-adenosyl-methionine-dependent methylation of C18:1 esterified in phospholipids[25]. In light of these results, we attribute the formation of PA from 1-acyl-2-hydroxy-*sn*-G3P and C18:1-CoA in reactions catalyzed by PlsB2 [Fig. 3b] to the non-enzymatic transmigration of the acyl chain in position *sn*-1 of LPA acceptors to position *sn*-2 thereby generating acceptor substrates that PlsB2 can use for the transfer of C18:1. This assumption is supported by the LC/MS analysis of the PA products of the reactions unambiguously showing that the transfer of C18:1 occurred at position *sn*-1 of each of the three *sn*-1-LPA substrates rather than *sn*-2 [Fig. S7]. The fact that PA formation was only clearly detectable when C18:1-CoA served as the acyl donor [Fig. 3b] indicates that the 2-acyl-G3P acyltransferase activity of PlsB2 is selective for C18:1-CoA as the donor substrate. This is in contrast with the G3P acyltransferase activity of the same enzyme which uses C18:0-CoA and C18:1-CoA with apparent comparable efficiencies [Fig. 3b].

## Effect of PlsM and PlsB2 expression on *E. coli* glycerolipid synthesis

Consistent with the major difference displayed by PlsMsmg and PlsC$_{coli}$ in terms of their ability to transfer acyl chains to position *sn*-2 of both G3P and 1-acyl-2-hydroxy-*sn*-G3P, or 1-acyl-2-hydroxy-*sn*-G3P only, phospholipid synthesis by *E. coli* membranes expressing *plsMsmg* was greatly impaired resulting in LPA build-up that accompanied a general decrease in [14C(U)]G3P incorporation in all phospholipid forms and a transient accumulation of monoacylglycerol (MAG)

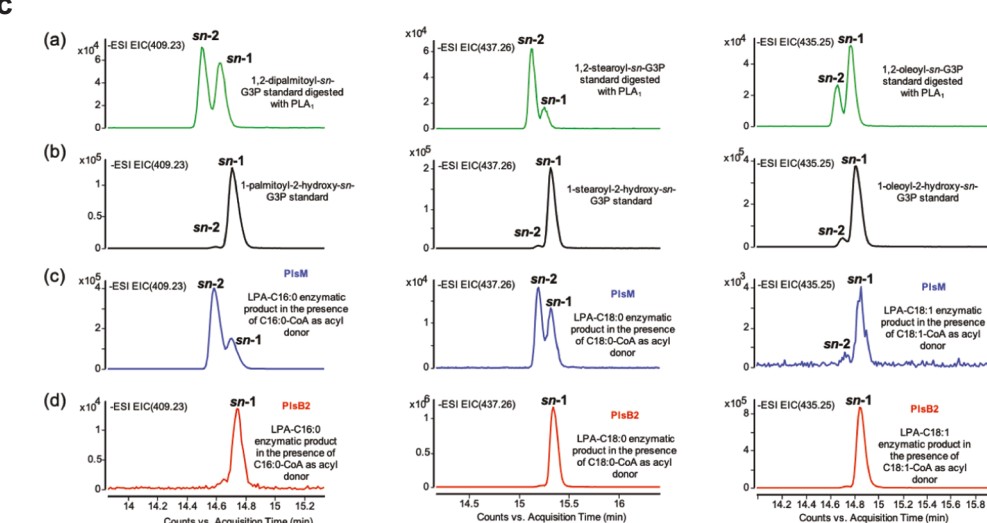

**b**

| Acceptor substrate | Acyl donor | % enzymatic conversion | | | |
|---|---|---|---|---|---|
| | | LPA product | No enzyme | PlsMsmg | PlsB2smg |
| G3P | C16:0-CoA | C16:0 | 0.13 ± 0.01 | 15 ± 0.01 | 0.35 ± 0.11 |
| G3P | C18:0-CoA | C18:0 | 0.12 ± 0.0 | 0.73 ± 0.16 | 14.6 ± 0.21 |
| G3P | C18:1-CoA | C18:1 | 0.09 ± 0.0 | 1.40 ± 0.25 | 9.04 ± 1.87 |
| Acceptor substrate | Acyl donor | PA product | No enzyme | PlsMsmg | PlsB2smg |
| G3P | C16:0-CoA | C16:0/C16:0 | 0.21 ± 0.03 | 6.90 ± 0.16* | 0.08 ± 0.00 |
| G3P | C18:0-CoA | C18:0/C18:0 | 0.03 ± 0.00 | 0.1 ± 0.02 | 0.05 ± 0.01 |
| G3P | C18:1-CoA | C18:1/C18:1 | 0.03 ± 0.00 | 0.01 ± 0.00 | 0.54 ± 0.02* |
| sn-1-LPA(C16:0) | C16:0-CoA | C16:0/C16:0 | 0.09 ± 0.01 | 98.2 ± 0.00 | 1.3 ± 0.20 |
| sn-1-LPA(C16:0) | C18:0-CoA | C16:0/C18:0 | 0.00 ± 0.00 | 51.9 ± 0.10 | 0.4 ± 0.30 |
| sn-1-LPA(C16:0) | C18:1-CoA | C16:0/C18:1 | 0.00 ± 0.00 | 30.1 ± 0.80 | 9.8 ± 0.00* |
| sn-1-LPA(C18:0) | C16:0-CoA | C18:0/C16:0 | 0.00 ± 0.00 | 96.2 ± 0.10 | 0.3 ± 0.10 |
| sn-1-LPA(C18:0) | C18:0-CoA | C18:0/C18:0 | 0.03 ± 0.00 | 55.2 ± 2.10 | 0.1 ± 0.10 |
| sn-1-LPA(C18:0) | C18:1-CoA | C18:0/C18:1 | 0.00 ± 0.00 | 30.2 ± 0.00 | 14.8 ± 2.60* |
| sn-1-LPA(C18:1) | C16:0-CoA | C18:1/C16:0 | 0.00 ± 0.00 | 70.3 ± 0.80 | 0.8 ± 0.10 |
| sn-1-LPA(C18:1) | C18:0-CoA | C18:1/C18:0 | 0.00 ± 0.00 | 5.8 ± 0.10 | 0.8 ± 0.10 |
| sn-1-LPA(C18:1) | C18:1-CoA | C18:1/C18:1 | 0.14 ± 0.02 | 8 ± 0.70 | 9.3 ± 0.30* |

[Fig. 6a]. We attribute this general decrease in phospholipid synthesis to the fact that the *E. coli* PlsB enzyme is unable to use the 1-hydroxy-2-acyl-*sn*-G3P products of PlsMsmg to generate PA[7]. MAG accumulation likely results from the dephosphorylation of unutilized 1-hydroxy-2-acyl-*sn*-G3Ps.

In contrast to the situation with *plsMsmg*, overexpression of *plsB2smg* stimulated phospholipid synthesis by *E. coli* membranes,

reflecting the enhanced ability of the recombinant strain to synthesize 1-acyl-2-hydroxy-*sn*-G3P that *E. coli* can utilize to produce glycerolipids [Fig. 6b].

## A model for PlsMtb and PlsB2tb substrates recognition
Recent success in protein structure prediction by artificial intelligence-based programs (in particular Alphafold[26,27]) made it possible to obtain

**Fig. 3 | G3P and LPA acyltransferase activity of PlsMsmg and PlsB2smg.**
**a** Schematic representation of PlsMsmg and PlsB2smg acyltransferase activities in the presence of G3P and various acyl-CoA donors. The arrows point to the positions at which acyl groups are transferred by the two enzymes. Preferred acyl donors in each reaction are in bold letters. **b** Percentage enzymatic conversion of G3P and various 1-acyl-2-hydroxy-*sn*-G3P (*sn*-1-LPAs) to their corresponding LPA and PA products by PlsMsmg and PlsB2smg. Peak areas for substrates and enzymatic products were obtained from the integration of extracted ion chromatograms (EICs) and used to calculate percentage substrate conversion. The LPA and PA products reported for the assays that used G3P as the acceptor substrate are from the same reactions. Values for LPA products include both *sn*-1 and *sn*-2-LPAs to take into consideration the spontaneous transmigration of acyl chains. Assays were run as described under Methods for 1 h at 37 °C. The results shown are the means ± standard deviations of duplicate assays and are representative of at least two independent experiments using different enzyme preparations. Asterisks denote

products resulting from the spontaneous transmigration of acyl chains between positions *sn*-1 and *sn*-2 as detailed in the text and Fig. S7. **c** G3P acyltransferase activity of PlsMsmg and PlsB2smg. (a) EICs showing LPAs generated from the enzymatic digestion of authentic phosphatidic acid (PA) standards (1,2-dipalmitoyl-*sn*-G3P, 1,2-stearoyl-*sn*-G3P and 1-palmitoyl-2-oleoyl-*sn*-G3P) with phospholipase A1 and analyzed by LC/MS. (b) EICs showing authentic standards of *sn*-1-palmitoyl-*sn*-2-hydroxy-G3P, *sn*-1-stearoyl-*sn*-2-hydroxy-G3P, and *sn*-1-oleoyl-*sn*-2-hydroxy-G3P analyzed by LC/MS. (c) EICs of PlsMsmg enzymatic products generated in the presence of G3P as acceptor substrate and C16:0-CoA, C18:0-CoA and C18:1-CoA as acyl donors. (d) EICs of PlsB2smg enzymatic products generated in the presence of G3P as acceptor substrate and C16:0-CoA, C18:0-CoA and C18:1-CoA as acyl donors. Non-radiolabeled PlsMsmg and PlsB2smg enzyme assays were run as described under Methods. The results shown for both enzymes are representative of at least two independent experiments using different enzyme preparations.

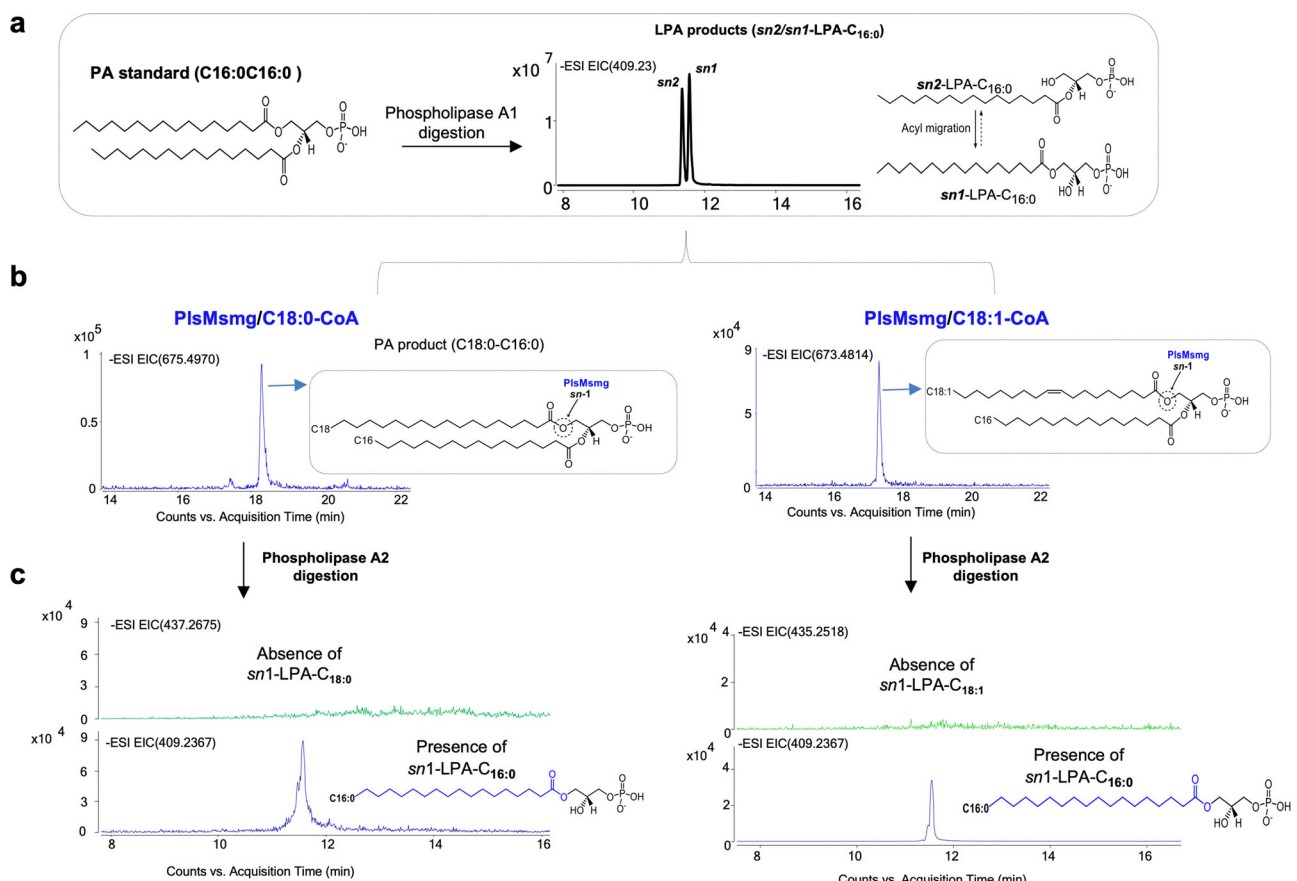

**Fig. 4 | PlsMsmg does not display 2-acyl-G3P acyltransferase activity.** The products of PlsMsmg acyltransferase reactions using 1-hydroxy-2-palmitoyl-G3P as the acceptor substrate and C18:1-CoA or C18:0-CoA as acyl donors were analyzed by LC/MS. **a** shows the generation of LPA acceptor substrates from 1,2-dipalmitoyl-*sn*-G3P upon phospholipase A1 digestion. Both *sn*-1 and *sn*-2 LPAs are found as a result of the spontaneous (non-enzymatic) transmigration of C16:0 from position *sn*-2 to

position *sn*-1. **b** Incubation of this LPA mixture with PlsMsmg in the presence of C18:1-CoA or C18:0-CoA yields PA products harboring C16:0/C18:0 or C16:0/C18:1 acyl chains. **c** Digestion of these PA products with phospholipase A2 solely yields *sn*-1 LPA products harboring a C16:0 chain indicating that PlsMsmg transferred C18:0 or C18:1 to the *sn*-2 position of 1-palmitoyl-2-hydroxy-G3P rather than to the *sn*-1 position of 1-hydroxy-2-palmitoyl-G3P.

reliable atomic models of PlsMtb (Rv2182c; Uniprot code O53516; 247 residues) and PlsB2tb (Rv2482c; Uniprot Code P9WI61; 789 residues) [Fig. 7 and S8]. Interestingly, a search for structural homologs using the DALI server[28] yielded (i) the 1-acyl-*sn*-G3P acyltransferase PlsC from *Thermotoga maritima* (*Tm*PlsC; PDB code 5KYM) as the closest homolog to PlsMtb (Z-score of 20.5; root mean squared deviation (r.m.s.d.) value of 2.2 Å for 187 aligned residues; 29% identity)[29] and (ii) the G3P acyltransferase from *Cucurbita moschata* (*Cm*GPAT; PDB codes 1K30 and 1IUQ) as the closest homologue to the acyltransferase domain of PlsB2tb (Z-score of 16.1; root mean squared deviation

(r.m.s.d.) value of 3.6 Å for 229 aligned residues; 10 % identity)[30–32]. *Cm*GPAT catalyzes the transfer of an acyl group from either acyl-ACPs or acyl-CoAs to the *sn*-1 position of G3P to yield 1-acyl-G3P. The structural superposition of the four acyltransferases shows a highly conserved catalytic HX$_4$D motif, where a histidine and an aspartic acid residue promote a charge-relay system to facilitate the nucleophilic attack, supporting a common catalytic mechanism (H41 and D46 in PlsMtb; H276 and D281 in PlsB2tb)[9,33,34].

To further investigate the mechanism of substrate recognition and specificity of the two mycobacterial enzymes, we performed

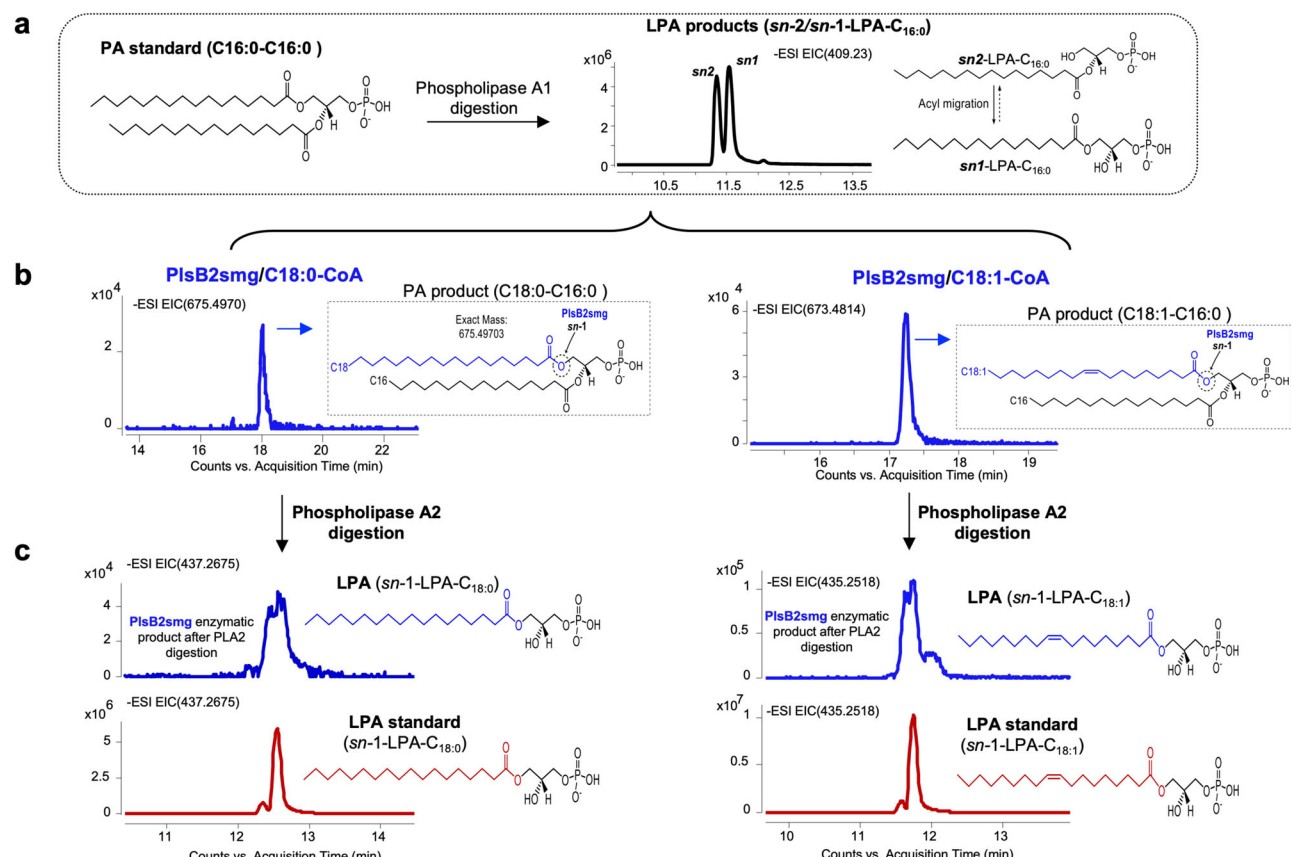

**Fig. 5 | PlsB2smg displays 2-acyl-G3P acyltransferase activity.** The products of PlsB2smg acyltransferase reactions using 1-hydroxy-2-palmitoyl-G3P as the acceptor substrate and C18:1-CoA or C18:0-CoA as acyl donors were analyzed by LC/MS. **a** The generation of *sn*-1 and *sn*-2 LPA acceptors from 1,2-dipalmitoyl-*sn*-G3P is as described in Fig. 4A. **b** Incubation of this LPA mixture with PlsB2smg in the presence of C18:1-CoA or C18:0-CoA yields PA products harboring C16:0/C18:0 or C16:0/C18:1 acyl chains. **c** Digestion of these PA products with phospholipase A2 yields *sn*-1 LPAs with C18:0 or C18:1 chains confirming that the PlsB2smg-mediated acyl transfer occurred at position *sn*-1 of 1-hydroxy-2-palmitoyl-G3P. *sn*-1 LPAs comigrate with authentic *sn*-1-LPA standards (red traces).

molecular docking calculations and molecular dynamic simulations on (i) PlsMtb in the presence of C16:0-CoA and G3P, and (ii) PlsB2tb in the presence of C18:0-CoA and G3P [Fig. 7 and S8]. In PlsMtb, the acyl chain perfectly accommodates into a hydrophobic groove supporting the specificity for C16:0 donor substrates. The 4-phosphopantetheinate moiety of C16:0-CoA is extended along a perpendicular main groove, with the adenosine 3′,5′-diphosphate (3′,5′-ADP) moiety of the ligand sticking out from the α/β globular core and exposed to the bulk solvent, as observed in other acyl-CoA modifying enzymes[34,35] [Fig. 7 and S8]. In PlsB2tb, the acyl chain of C18:0-CoA is placed into a long hydrophobic tunnel likely to facilitate the recognition of longer acyl chain length substrates (C18) over that of shorter ones (C16). The 4-phosphopantetheinate and 3′,5′-ADP moieties accommodate in a perpendicular main groove as observed in PlsMtb [Fig. 7 and S8]. It is worth noting that both main grooves are decorated with surface-exposed, positively charged, residues and hydrophobic residues compatible with membrane association[36]. In both PlsMtb and PlsB2tb, the G3P acceptor substrate is in a region at the end of the corresponding main grooves, mainly decorated by different polar lateral chains [Fig. 7 and S8]. The phosphate group is basically anchored by the surrounding residues in order to orientate the OH in the correct position toward the scissile linkage of the donor substrate with the catalytic residues favoring the transfer of an acyl group in *sn*-2 or *sn*-1 in PlsMtb and PlsB2, respectively. Due to the lack of a second hydrophobic cavity which could accommodate the acyl chain of LPAs, the models support a positioning of the acyl chains in the LPA acceptor substrates away from the protein surface, in proximity to the lipid bilayer.

## Discussion

The two-fold acylation of the *sn*-1 and *sn*-2 positions of G3P by G3P-acyltransferases and LPA-acyltransferases results in the formation of PA, the common precursor to glycerophospholipids and triglycerides. Despite the universal conservation of this process in prokaryotes, it is becoming apparent that bacteria have evolved variations in the biosynthetic steps leading to PA, reflecting differences in their physiology and lifestyle[5,9,10]. In this regard, mycobacteria have evolved at least four remarkable variations of their own. Firstly, unlike most bacteria but similar to the situation encountered in archaea, eukarya and Xanthomanadales, mycobacteria are devoid of a glycerophosphate acyltransferase PlsX/PlsY system and instead exclusively rely on a PlsB/PlsC-like pathway. Secondly, mycobacterial glycerolipid acyltransferases require acyl-CoAs rather than acyl-ACPs as acyl donors[7,8] (this study). Thirdly is the unusual positional distribution of fatty acids in mycobacterial glycerolipids wherein unsaturated or branched oleyl (C18:1) or tuberculostearoyl (C19) substituents principally esterify position 1, and palmitoyl (C16:0) principally occupy position 2[3]. The results of our combined cell-free assays, whole cell-based experiments and structural modeling provide an enzymatic rationale for this structural oddity by implicating the essential acyltransferase PlsM in the selective transfer of C16:0 to position *sn*-2, and PlsB2 in the preferential transfer of C18:1 (or C18:0 depending on the acceptor substrate) to position *sn*-1. Fourth and lastly, the characterization of PlsM as an essential PlsC-type enzyme capable of catalyzing the acylation of position *sn*-2 of both G3P and 1-acyl-G3P, while PlsB2 catalyzes the acylation of position *sn*-1 of both G3P and 2-acyl-G3P supports the existence of two parallel PA biosynthetic pathways in mycobacteria, a

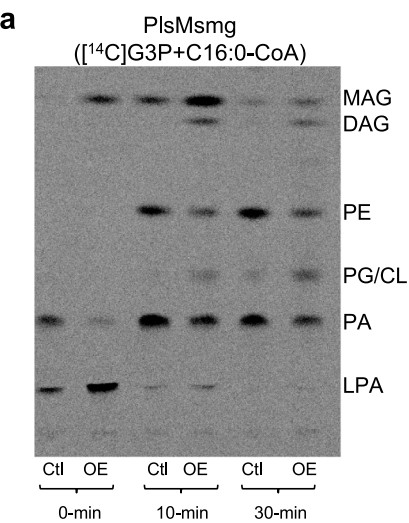

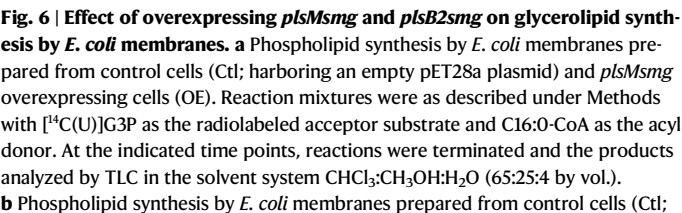

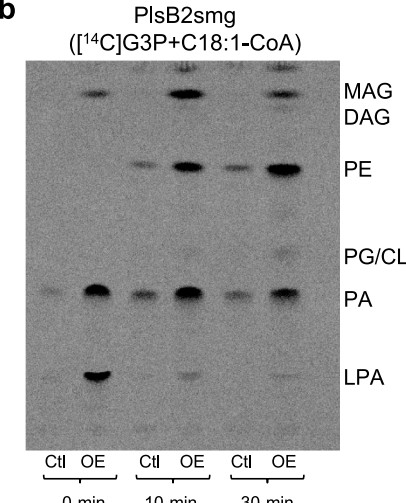

**Fig. 6 | Effect of overexpressing *plsMsmg* and *plsB2smg* on glycerolipid synthesis by *E. coli* membranes. a** Phospholipid synthesis by *E. coli* membranes prepared from control cells (Ctl; harboring an empty pET28a plasmid) and *plsMsmg* overexpressing cells (OE). Reaction mixtures were as described under Methods with [$^{14}$C(U)]G3P as the radiolabeled acceptor substrate and C16:0-CoA as the acyl donor. At the indicated time points, reactions were terminated and the products analyzed by TLC in the solvent system CHCl$_3$:CH$_3$OH:H$_2$O (65:25:4 by vol.). **b** Phospholipid synthesis by *E. coli* membranes prepared from control cells (Ctl; harboring an empty pET14b plasmid) and *plsB2smg* overexpressing cells (OE). Reaction mixtures were as described under Methods with [$^{14}$C(U)]G3P as the radiolabeled acceptor substrate and C18:1-CoA as the acyl donor. At the indicated time points, reactions were terminated and the products analyzed as described above. The results shown are representative of two (PlsB2) to three (PlsM) independent experiments. LPA, lysophosphatidic acid; PA, phosphatidic acid; CL, cardiolipin; PG, phosphatidylglycerol; PE, phosphatidylethanolamine; MAG, monoacylglycerol; DAG, diacylglycerol. Source data are provided as a Source Data file.

classical one wherein the acylation of the *sn*-1 position of G3P precedes that of *sn*−2 and another wherein acylations proceed in the reverse order[7]. The proposed steps leading to the biosynthesis of PA in mycobacteria are summarized in Fig. 8.

Which of the two branches leading to the synthesis PA prevails in intact mycobacterial cells cannot be readily inferred from our studies and, as pointed out earlier, may depend on the conditions under which mycobacteria are grown[3,7]. Cell-free assays using membranes from *M. butyricum*[7] and our own assays using *Msmg* membranes [Fig. 2e] suggest the 2-acyl-G3P acyltransferase system (colored in red on Fig. 8) to be the dominant one. Importantly, based on the marked preference of PlsB2 for C18:1-CoA when *sn*-2-LPAs serve as acceptors [Fig. 3b], the 2-acyl-G3P acyltransferase system also stands out as the most compatible with the positional distribution of acyl chains in mycobacterial glycerolipids. The fact that PlsB2 is not essential for *Mtb* growth[14] indicates that either the 2-acyl-G3P branch leading to PA synthesis is not essential for growth or that there is some functional redundancy between PlsB2 and the second PlsB homolog of *Mtb*, PlsB1[6] [Table S1]. PlsM, in contrast, is an essential enzyme indicating that none of the other five PlsC homologs encoded by the *Mtb* genome are able to qualitatively or quantitatively compensate for its activity in either branch of the PA biosynthetic pathway. This is supported by the fact that silencing *plsM* in *Msmg* not only led to a dramatic decrease in phospholipid synthesis [Fig. 1] but also to a marked decrease in the proportion of glycerolipids esterified with saturated acyl chains indicative of the lesser ability of other PlsCs to transfer C16:0 to the *sn*-2 position [Fig. 1g]. Furthermore, cell-free assays aimed at analyzing the 2-acyl-G3P acyltransferase branch of glycerolipid synthesis in *Msmg* extracts indicated that the ability of *Msmg* membranes to transfer C16:0 to position *sn*-2 of G3P is lost in the absence of PlsM (i.e., in *MsmgΔplsM*/pMVGH1-*plsCcoli*) [Fig. 2e].

G3P acyltransferases with *sn*-2 regiospecificity have previously been reported in plants but, until now, were thought to be absent from animals, fungi and microorganisms (including algae)[37,38]. To differentiate them from the traditional *sn*−1 G3P acyltransferases (Enzyme Commission [EC] 2.3.1.15) involved in the de novo synthesis of membrane and storage lipids in bacteria, fungi, animals and plants, land-

plant-specific *sn*-2 G3P acyltransferases were classified in a family of their own, EC 2.3.1.198. PlsM is thus the first example of an *sn*-2 G3P acyltransferase outside the plant kingdom. As indicated above, it is also unique in its ability to acylate position *sn*-2 of both G3P and 1-acyl-G3P substrates. Furthermore, unlike plant *sn*-2 G3P acyltransferases which have evolved to provide precursors for cutin and suberin biosynthesis rather than membrane and storage lipid biosynthesis, PlsM is essential for the biosynthesis of all major forms of mycobacterial glycero(phospho)lipids. PlsB2 is also unique in its ability to transfer C18:0 and C18:1 to the *sn*-1 position of both G3P and 2-acyl-G3P substrates.

The physiological significance of the unusual distribution of fatty acids in mycobacterial glycerolipids is unknown. The positioning of saturated and unsaturated fatty acids in membrane phospholipids may affect their physicochemical properties[39]. Moreover, the position of fatty acids in prokaryotic and eukaryotic systems has been shown to affect their turnover and the susceptibility of phospholipids to phospholipases[3,7,40]. Another way the positional distribution of fatty acids in phospholipids affects mycobacterial physiology concerns tuberculostearic acid synthesis. Indeed, an accumulation of oleic acid at the expense of tuberculostearic acid was noted both in the *MsmgΔplsM* mutant rescued with *plsCcoli* and the conditional *MsmgΔplsM*/pSETetR-*plsMsmg* mutant grown under non-permissive conditions [Table S2; Fig. 2d]. Since tuberculostearic acid is known to arise from the methylation of oleic acid esterified in phospholipids[25,41], these results most likely reflect the untoward position of oleic acid in the phospholipids produced by the recombinant strains. Tuberculostearic acid-containing phospholipids play a critical role in the physiology of mycobacteria as recently evidenced by their involvement in the formation of intracellular membrane domains[41]. Finally, similar to the situation in plants where *sn*-1 and *sn*-2 G3P acyltransferases provide LPA and monoglyceride precursors that are used in different pathways[37,38], one cannot exclude that some of *Mtb*'s PlsC homologs contribute compositionally different LPA and PA precursors than those provided by PlsM that end up being used in the synthesis of different discrete end products.

Clearly the essentiality and distinctive substrate specificity of PlsM make this enzyme a novel therapeutic target of interest in the context of

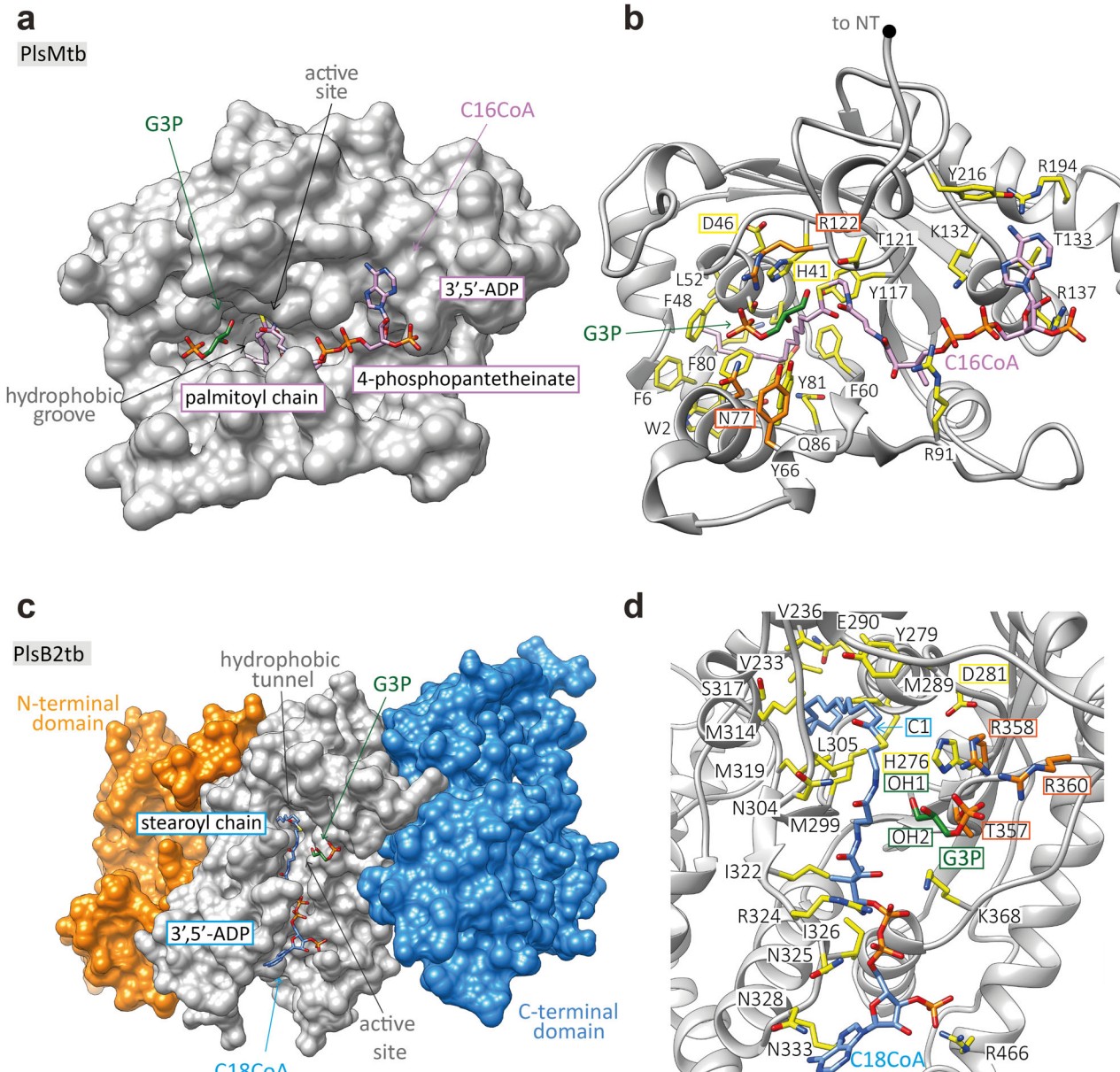

**Fig. 7 | A model for PlsMtb and PlsB2tb substrates recognition. a** Surface representation of the predicted PlsMtb 3D structure, showing the palmitoyl moiety of C16:0-CoA deeply buried into a hydrophobic groove. This hydrophobic pocket runs perpendicular with respect to a main groove where the CoA moiety is located. The 4-phosphopantetheinate moiety of C16:0-CoA is placed in the hydrophobic groove entrance, nearby the catalytic site, with the adenosine 3′,5′-ADP moiety extended along the protein surface. The G3P acceptor substrate is positioned in the opposite site of the CoA group, in a binding pocket mainly decorated by polar residues. **b** The palmitoyl moiety of C16:0-CoA makes interactions with W2, H41, S47, F48, P51, L52, F60, F80, Y81, S84, Q86 along the groove's walls and with F6, K7 and Y3 in the bottom, supporting PlsMtb's specificity for C16-length donor substrates. The 4-phosphopantetheinate moiety of C16:0-CoA is placed nearby the catalytic residues H41 and D46 and stabilized by Y117 with the adenosine 3′,5′-ADP moiety making interactions with residues R91, T121, R137, T133, R194, Y216 and K132. The G3P is stabilized by the R122, Y66 and N77 residues. **c** Surface

representation of the predicted PlsB2tb 3D structure, showing the stearoyl moiety of C18:0-CoA deeply buried into a hydrophobic tunnel of the acyltransferase domain. This hydrophobic tunnel runs perpendicular with respect to a main groove where the CoA moiety is located. The 4-phosphopantetheinate moiety of C18:0-CoA is placed in the hydrophobic groove entrance, nearby the catalytic site, with the adenosine 3′,5′-ADP moiety extended along the protein surface. The G3P acceptor substrate is positioned in the opposite site of the CoA group, in a binding pocket mainly decorated by polar residues. **d** The stearoyl moiety of C18:0-CoA makes interactions with E290, M289, V233, P286, S317, M314, M319, M299, V236, V283 and Y279 along the groove's walls. The 4-phosphopantetheinate moiety of C18:0-CoA is placed nearby the catalytic residues H276 and D281 and further stabilized by L305, N304 and I322, with the adenosine 3′,5′-ADP moiety making interactions with residues N325, I326, K368, N328, K333, R466 and R324. G3P is stabilized by the R358, R360 and T357 residues.

the development of much-needed new TB and nontuberculous mycobacterial drugs. The possibility of purifying PlsMsmg from *E. coli* has opened the way to crystallographic studies which should not only allow us to gain deeper insight into the molecular bases of the substrate selectivity of this enzyme but also facilitate the screening and rational development of inhibitors of glycerolipid synthesis in mycobacteria.

## Methods

### Bacterial strains and culture conditions

*M. smegmatis* mc²155 (*Msmg*) was grown in Middlebrook 7H9 medium (Difco) supplemented with 10% albumin-dextrose-catalase or oleic acid-albumin-dextrose-catalase supplements (BD Biosciences) and 0.05% tyloxapol or 0.05% Tween 80, on Luria Bertani (LB) agar (pH 7.5)

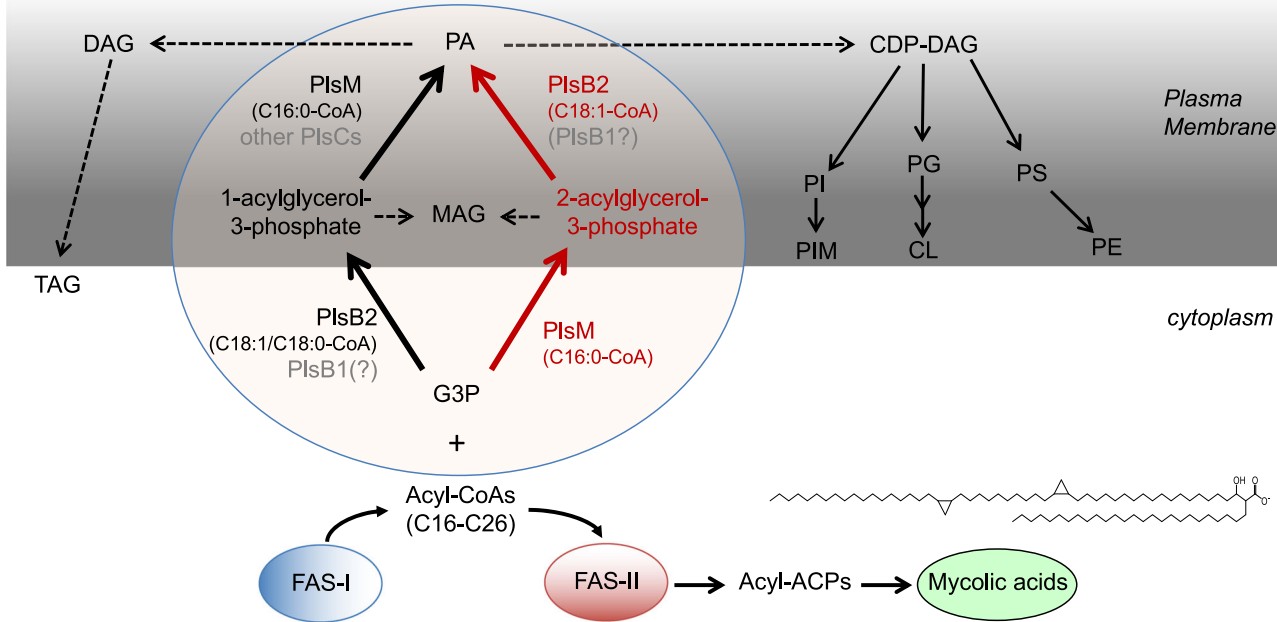

**Fig. 8 | Proposed pathway for glycerolipid synthesis in mycobacteria.** The proposed steps leading to the biosynthesis of phosphatidic acid (PA) and derived glycerolipids in *Mtb* based on earlier biochemical studies[7] and the present work are shown. The acyl-CoA products of FAS-I serve as the acyl donors in the biosynthesis of PA from glycerol-3-phosphate (G3P). The initial substrates of FAS-II are medium length (C16–C26) keto-acyl-ACP resulting from the condensation of the acyl-CoA products of FAS-I with malonyl-ACP. The processive addition of multiple malonate units to these precursors leads to the elongation of the meromycolate chain (C48–C54) of mycolic acids. The prototypical structure of an *Mtb* alpha-mycolate is shown. Glycerophospholipids and di- and tri-acylglycerol (DAG and TAG) arise from the central intermediate, PA. The catalytic activity of the enzymes in gray font has not been confirmed. "Other PlsCs" refers, in particular, to *Mtb* PlsC homologs Rv2483c and Rv3816c which are conserved in other *Mycobacterium* species, including *M. leprae* [Table S1]. Other putative PlsC candidates otherwise include Rv3814c, Rv3815c and Rv3026c in *Mtb*. PS, phosphatidylserine; PI, phosphatidyl-*myo*-inositol; PIM, phosphatidylinositol mannosides; TAG, triglycerides. Other lipid abbreviations are as in Fig. 6.

(Bactotryptone, 10 g/L, Bacto™ yeast extract, 5 g/L, NaCl, 5 g/L) (Becton Dickinson, Sparks, MD) or in Sauton's minimal medium. *E. coli* DH5α, the strain used for cloning, and *E. coli* BL21(DE3), the strain used for *MSMEG_4248* (*plsMsmg*) and *MSMEG_4703* (*plsB2smg*) expression, were propagated in LB broth. Where indicated, ampicillin (Amp), kanamycin (Kan), hygromycin (Hyg) were added to final concentrations of 100 µg/mL, 20 µg/mL and 50 µg/mL, respectively. When required, 10% sucrose was added to the solid medium.

### Knock-out of *MSMEG_4248* (*plsMsmg*) in *Msmg*

A two-step procedure employing the counterselectable marker *sacB*[42] was used to achieve allelic replacement at the *plsMsmg* (*MSMEG_4248*) locus of *Msmg* mc²155. Briefly, the *M. smegmatis plsM* gene and flanking regions was PCR-amplified from *Msmg* mc²155 genomic DNA using the primers plsMsmg.1 (5'- tataatctagaaggtgtggccggtatgac–3') and plsMsmg.2 (5'- tataagaattctgatctatctcgaaccgatc –3') and a disrupted allele, *plsM::kan*, was obtained by replacing 282 bp of the coding sequence of this gene flanked by PstI and NruI restriction sites with the kanamycin resistance cassette from pUC4K (GE Healthcare). *plsMsmg::kan* was cloned into pPR27-*xylE*[42] yielding the construct used for allelic replacement. pSETetR-*plsMsmg* and pSETetR-*plsMtb* are episomal rescue plasmids[43] in which the full-length *plsMsmg* and *plsMtb* genes were placed under control of a tetracycline-inducible (TET-ON) promoter. pMVGH1-*plsCcoli* and pMVGH1-*plsCsubtilis* were obtained by cloning the entire coding sequences of the *plsC* genes from *E. coli* and *Bacillus subtilis* under control of the *hsp60* promoter in the mycobacterial expression plasmid pMVGH1[43]. pMVGH1 harbors an hygromycin resistance cassette and allows for the production of C-terminal hexahistidine-tagged proteins. The primer sequences used in the construction of the rescue plasmids are available upon request. Allelic replacement at the *plsM* locus of *Msmg* was confirmed by PCR using different primer combinations.

### RNA preparation, reverse transcription and RT-qPCR

RNA was extracted from 5-mL cultures of *Msmg* strains expressing *plsCcoli* or *plsCsubtilis* from pMVGH1 grown in 7H9-ADC-tyloxapol medium to an OD$_{600}$ of ~0.2 using the Direct-zol™ RNA Miniprep kit (Zymo Research) per the manufacturer's instructions. Reverse transcription reactions were carried out using the Superscript IV First-Strand Synthesis System (ThermoFisher). RT-qPCRs were run using SsoAdvanced Universal SYBR Green Supermix (Bio-Rad) and analyzed on a CFX96 real-time PCR machine (Biorad). PCR conditions: 98 °C (30 s enzyme activation), followed by 40 cycles of 98 °C (10 s; denaturation) and 60 °C (30 s; annealing/extension). Mock reactions (no reverse transcription) were done on each RNA sample to rule out DNA contamination. The target cDNA was normalized internally to the *sigA* cDNA levels in the same sample. The following primers were used: *plsCcoli* fwd (5'-cgctgtttggcctgaaagtt –3'), *plsCcoli* rev (5'-tcgatgctgtcaccatgtca-3'); *plsCsubtilis* fwd (5'-tcgtcggtatcttcccaagc-3'), *plsCsubtilis* rev (5'-tgatatgcagcgggga-caag-3'), and *sigA_fwd* (5'-cttgaggtgaccgacgatct-3'), *sigA_rev* (5'-gagttccaggtcggtgtctt –3').

### Production and purification of PlsMsmg and PlsB2smg in *E. coli*

A recombinant form of PlsMsmg was produced in *E. coli* BL21(DE3) using the pET28a expression system (Novagen, Madison, WI). Following an overnight-induction with 1 mM IPTG at 16 °C in LB-Kan broth, *E. coli* BL21(DE3) cells transformed with pET28a-*plsMsmg* were harvested, washed and resuspended in lysis buffer consisting of 50 mM Tris-HCl (pH 7.5), 1 M NaCl, 10% glycerol, 2 mM CHAPS, 30 mM imidazole and protease inhibitors (Complete EDTA-free, Roche). Cells were disrupted using a French Press and the lysate was further incubated with 20 mM CHAPS for 1 h at 4 °C prior to centrifugation at 20,000 x g for 40 min to optimize the recovery of PlsMsmg from *E. coli* membranes. PlsMsmg was then purified by applying the resulting cell lysate to a HisTrap

Chelating column (GE HealthCare) equilibrated in buffer A (50 mM Tris-HCl pH 7.5, 1 M NaCl, 2 mM CHAPS) containing 30 mM imidazole. The column was washed with buffer A plus 50 mM imidazole until no absorbance at 280 nm was detected. Elution was performed with a linear gradient of 50–250 mM imidazole in 10 ml buffer A. Fractions containing PlsMsmg were pooled and kept in buffer consisting of 500 mM NaCl, 50 mM Tris-HCl (pH 7.5), 30 mM imidazole and 10% glycerol buffer at −80 °C until used in enzyme assays. Membranes from *E. coli* BL21(DE3) cells harboring either an empty pET28a plasmid or pET28a-*plsMsmg* were prepared in 50 mM Tris-HCl (pH 7.4) and also stored at −80 °C until used in enzyme assays.

A recombinant form of PlsB2smg was produced in *E. coli* BL21(DE3) using the pET14b expression system (Novagen, Madison, WI). Following induction with 1 mM IPTG at 16 °C in LB-Amp broth, cells were harvested, washed and resuspended in lysis buffer consisting of 50 mM Tris-HCl (pH 7.5), 1 M NaCl, 10% glycerol, 1 mM CHAPS, and protease inhibitor (0.2 mM 4-(2-aminoethyl)benzenesulfonyl fluoride hydrochloride). Cells were disrupted by sonication and the lysate was further incubated for 1 h at 4 °C under shaking prior to centrifugation as above to optimize the recovery of PlsB2smg from *E. coli* membranes. An equal volume of lysis buffer was then added to the supernatant and loaded onto the Ni Excel Sepharose column (Cytiva). The column was washed with 10% lysis buffer containing 5 mM imidazole. Elution was performed with a linear gradient of 50-100 mM imidazole after which eluted fractions were pooled and concentrated using a Vivaspin concentrator MWCO 50 kDa (Sartorius). Concentrated samples containing PlsB2smg were dialyzed overnight at 4 °C in a buffer containing 50 mM Tris pH 7.5, 500 mM NaCl, 30 mM imidazole, and 10% glycerol, flash-frozen in liquid nitrogen and stored at −80 °C until used in enzyme assays.

### Whole cell radiolabeling experiments
Radiolabeling of whole *Msmg* cultures (WT *Msmg* mc²155, *Msmg*Δ*plsMsmg*/pSETetR-*plsMtb* and *Msmg*Δ*plsMsmg*/pMVGH1-*plsC*$_{coli}$ clones # 32 and 34) grown to an OD$_{600}$ of 0.8 with [1,2-¹⁴C]acetic acid (0.5 μCi/ml; specific activity, 54.3 Ci/mol, Perkin Elmer) was performed in 7H9-ADC-tyloxapol medium for 8 h at 37 °C with shaking.

### Lipid extraction and preparation of fatty acid methyl esters
Total lipids extraction from bacterial cells and preparation of fatty acid methyl esters from extractable lipids followed earlier procedures[44].

### Analysis of lipids and fatty acids from whole cells
Cold and [1,2-¹⁴C] acetic acid-derived lipids and fatty acid methyl esters were analyzed by TLC on aluminum-backed silica gel 60-precoated plates F$_{254}$ (E. Merck). TLC plates were revealed by spraying with cupric sulfate (10% in an 8% phosphoric acid solution) and heating. Radiolabeled products were analyzed using a PhosphorImager (Typhoon, GE Healthcare).

Alternatively, lipids were analyzed by LC/MS in both positive and negative mode on a high-resolution Agilent 6220 TOF mass spectrometer interfaced to a LC as described[45]. For quantification purposes, 20 μl of 1 mg/mL pre-mixed deuterium-labeled internal glycerophospholipid standard (SPLASH LIPIDOMIX Mass Spec Standard from Avanti, Polar Lipids, USA) was added to the total lipids prior to injection. Data files were analyzed with Agilent's Mass hunter workstation software (Version B.02.00, build 2.0.197.0) to identify compounds using "molecular feature extractor". Agilent's mass profiler program was used to compare the various lipids present in analogous control and recombinant strains. Most lipids were identified using a database of *Mtb* lipids developed in-house[45]. Peak areas of the extracted ion chromatograms (EICs) for monoacylglycerides, DAG, and TAG were obtained in positive ionization mode to quantify glycerophospholipids compositions. EICs for LPE, PA, PS, PG, PE, CL, PI, Ac$_1$PIM$_2$, and Ac$_2$PIM$_2$ were obtained in negative ionization mode. Peak areas of EICs for

deuterium labeled 15:0-18:1(d7) PI were used to quantify CL, Ac$_1$PIM$_2$, and Ac$_2$PIM$_2$. Peak areas of EICs were used to calculate the relative abundance of saturated *vs.* unsaturated glycerophospholipids. See Supplementary File 1 for more details about the LC/MS procedures used in this study.

Fatty acid methyl esters prepared from extractable lipids were analyzed using a CP 3800 gas chromatograph (Varian) equipped with an MS320 mass spectrometer in the electron impact mode and scanning from $m/z$ 50 to $m/z$ 1000 over 0.5 s. Helium was used as the carrier gas with a flow rate of 1 mL/min. The samples were run on an Agilent VF-5ms column (30 m x 0.25 mm i.d.). The injector (splitless mode) was set at 250 °C. The oven temperature was held at 50 °C for 1.5 min, programmed at 30 °C/min to 120 °C and then programmed at 10 °C/min to 330 °C followed by a 5 min hold. Data analyses were carried out on a Varian WS data station.

### Synthesis of C16:0-ACP
The C16-AcpM was prepared enzymatically from palmitic acid (Sigma-Aldrich, St. Louis, MO) and *Mtb* holo-AcpM in the presence of *E. coli* acyl-ACP synthase as previously reported[46].

### Radiolabeled PlsM acyltransferase assays
Radiolabeled G3P acyltransferase assay mixtures contained 0.3–0.5 μCi [¹⁴C(U)]G3P (specific activity 150 mCi/mmol, Perkin Elmer), 150 mM NaCl, 1 mg/mL BSA, 2.5–40 μM acyl donor (palmitoyl-CoA [Avanti Polar Lipids, Inc.] or palmitoyl-ACP), purified PlsMsmg or *E. coli* membranes expressing or not *plsMsmg* (440 μg of proteins), and 50 mM Tris-HCl (pH 7.4) buffer in a final volume of 100 μL. After incubation at 37˚C, the reactions were stopped by addition of 3 mL of CHCl$_3$/CH$_3$OH (2:1) and 0.4 mL 0.9% NaCl.

### Non-radiolabeled acyltransferase assays with purified enzymes
Cold G3P acyltransferase assay mixtures contained 200 μM G3P (Sigma), 100 μM of palmitoyl-CoA, stearoyl-CoA or oleoyl-CoA (Avanti Polar Lipids, Inc), purified PlsMsmg (20 μg) or PlsB2smg (50 μg), and 50 mM Tris-HCl (pH 7.4) buffer containing 150 mM NaCl in a final volume of 100 μL. In 1-acyl-G3P acyltransferase assays, 100 μM 1-acyl-2-hydroxy-G3P [Avanti Polar Lipids, Inc] replaced G3P. In 2-acyl-G3P acyltransferase assays, 1-hydroxy-2-palmitoyl-G3P generated from commercial 1,2-dipalmitoyl-*sn*-G3P (1 mg) upon phospholipase A1 digestion (see next section) replaced G3P. Ten percent of the products resulting from the phospholipase A1 digestion and 100 μM stearoyl-CoA or oleoyl-CoA were used as acceptor and donor substrates, respectively in PlsB2smg assays. To confirm the position at which acyl chains were transferred by PlsB2smg, the PA products of the PlsB2smg reactions were digested with phospholipase A2 (10 μg) in 50 mM Tris-HCl buffer (pH 7) containing 150 mM NaCl and the resulting products analyzed by LC/MS as described below. Unless otherwise indicated, all enzyme reactions were incubated for 1 h at 37 °C. Reactions were stopped by addition of a freshly prepared 1:1 (v/v) 0.1 M HCl and CH$_3$OH solution (1.6 mL)[47], vortexed for 30 s and kept on ice. Prechilled CHCl$_3$ (0.8 mL) containing 1 μg of deuterated internal standard (1-pentadecanoyl-2-oleoyl(d7)-*sn*-G3P) was then added to all the tubes, vortexed for 30 s and centrifuged at 2000 x *g* for 5 min at 4 °C. The lower organic layer was dried under nitrogen and resuspended in 200 μL of prechilled acidic methanol (pH 4)[23] prior to LC/MS analysis. Several studies have reported on the rapid, non-enzymatic, isomerization of *sn*-2 acyl isomers of LPA to the more stable *sn*-1 acyl isomers[23,24]. The ratio of *sn*-1 acyl isomer to *sn*-2 acyl isomer has been shown to be ~9:1 at equilibrium in aqueous solution. Solutions at or above physiological pH, high temperatures and the presence of BSA in reaction mixtures tended to accelerate acyl migration. Precautions were thus taken throughout our assays and sample preparation and analysis to mitigate the issue by using recommended protocols[23,24].

### Preparation of 1-hydroxy-2-acyl-G3P standards

Synthetic PA standards (1,2-dipalmitoyl-*sn*-G3P, 1,2-stearoyl-*sn*-G3P, and 1-palmitoyl-2-oleoyl-*sn*-G3P) were purchased from Avanti Polar Lipids, and Phospholipase A1 from Sigma (L3295, A1 activity >10 KLU/G). Phospholipase A1 was diluted (1:20) in 50 mM Tris-HCl pH 7.4 containing 150 mM NaCl and incubated with PA standards at a final concentration of 100 μM for 30 min at 37 °C. The digested samples were extracted as described above for non-radiolabeled G3P acyltransferase assays and analyzed by LC/MS.

### LC/MS analysis of enzymatic products

LC separation and mass spectrometry analyses of PlsMsmg and PlsB2smg enzymatic products were performed on an Agilent 6520 A time-of-flight (TOF) mass spectrometer equipped with an Agilent 1312B binary pump. An Agilent Poroshell 120EC-C8 column (100 mm × 2.1 mm; 2.7 μM) was used to separate analytes with a flow rate of 0.30 mL/min using solvent A (0.1% formic acid in water) and solvent B (0.1% formic acid in acetonitrile). Typically, analytes were injected in 5% solvent B and LC separation was performed using the following gradient conditions: 0–1 min (5% B), 1-3 min (20% B), 3-6 min (35% B), 6-10 min (98% B), 10–29 min (98% B), and 29-30 min (5% B). Agilent 6520 A Q-TOF comprises electrospray ionization/atmospheric pressure chemical ionization (ESI/APCI) and a dual ESI source which was operated in both negative and positive ion modes. The mass spectra were recorded at a rate of 1.02 spectra/s with a data acquisition range of $m/z$ 250-3200 Da. Data processing was carried out using Mass Hunter Workstation Software Qualitative Analysis (version B.07.00).

### G3P acyltransferase assays using *Msmg* cell-free extracts

To determine and compare the nature of the LPA products generated by cell-free extracts from *Msmg*/pMVGH1 and *MsmgΔplsMsmg*/pMVGH1-*plsC$_{coli}$*, 400 μg of membranes prepared from each of the two strains in Tris-HCl buffer (pH 7) as described previously[48] were incubated at 37 °C for 1 min with 2 mM *sn*-[UL-$^{13}$C$_3$]-G3P (Omicron Biochemicals, Inc.) and 100 μM of stearoyl-CoA or oleoyl-CoA (Avanti Polar Lipids, Inc) in 50 mM Tris-HCl (pH 7.4) containing 150 mM NaCl in a total volume of 100 μL. Reactions were stopped by the addition of freshly prepared 1:1 (v/v) 0.1 M HCl (0.8 mL) and prechilled CHCl$_3$ (0.4 mL). The reaction products were centrifuged at 3500 rpm for 5 min, and the lower organic layer was dried under nitrogen and stored at −20 °C until LC/MS analysis.

### Sliding motility assay

Motility plates consisted of M63 medium containing 0.2% glucose with or without added Tween 80 (0.05%) (plus antibiotics for the recombinant strain) solidified with 0.3% ultrapure agarose[49]. *Msmg* control and test strains were inoculated at the center of the motility plates and incubated at 37 °C under a humid 5% CO$_2$ atmosphere for up to 25 days.

### Drug susceptibility testing

MIC values of various antibiotics against *Msmg* WT and recombinant strains grown to mid-log phase (OD600 nm ~0.8) were determined in 7H9-ADC-tyloxapol (0.05%) at 37 °C in 96-well microtiter plates using the colorimetric resazurin microtiter assay[50].

### Molecular docking calculations and MD simulations

PlsMtb and PlsB2tb structure model predictions were computationally calculated by Alphafold software[26,27]. Ligand docking was performed using AutoDock Vina employing standard parameters[51]. The simulations were carried out with AMBER 20 package implemented with ff14SB and GLYCAM06 force fields[52]. The system was neutralized and the complex was buried in a water box with a 10 Å of TIP3P water molecules. A two-stage geometry optimization approach was performed. The first stage minimizes only the positions of solvent molecules, and the second stage is an unrestrained minimization of all the atoms in the simulation cell. The system was heated from 0 to 300 K under the constant pressure of 1 atm and periodic boundary conditions. Harmonic restraints of 30 kcal/mol were applied to the solute, and the Andersen temperature coupling scheme was used to control and equalize the temperature. Long-range electrostatic effects were modelled using the particle-mesh-Ewald method. An 8 Å cut-off was applied to Lennard-Jones interactions. Each system was equilibrated for 2 ns with a 2-fs time step at a constant volume and temperature of 300 K. Production trajectories were then run for additional 1 μs under the same simulation conditions. The obtained coordinate trajectories and data files were analysed with cpptraj software in AMBER software package. Structure based homologs search and Z-score values were produced by using DALI and molecular modeling visualization was performed using USCF Chimera[53].

### Statistics and reproducibility

Unless otherwise indicated in the figure legends, data are expressed as the mean ± SD values from duplicate or triplicate assays ($n = 2$–3 biologically independent samples). An unpaired Student's $t$-test was used as the statistical test in the different experiments as indicated in the figure legends. Calculations were performed using Graphpad Prism version 9.5.1 for Windows (San Diego, CA, USA). Asterisks on the graphs denote cut-off $P$-values as indicated in the legends. No statistical method was used to predetermine sample size. No data were excluded from the analyses. The experiments were not randomized and the investigators were not blinded to allocation during experiments. Investigator blinding was performed for analysis of LC/MS and GC/MS data.

### Availability of biological materials

Biological materials generated in the context of this work such as recombinant bacterial strains and expression plasmids for the production of PlsM and PlsB2 will be made available upon request to the corresponding author and may be subject to the terms of a Material Transfer Agreement.

### Reporting summary

Further information on research design is available in the Nature Portfolio Reporting Summary linked to this article.

## Data availability

The data generated in this study, including the processed LC/MS data, are available within the paper, its supplementary information files and Source Data file. Raw LC/MS data are available as Agilent Mass Hunter workstation files from the corresponding author upon reasonable request. Source data are provided with this paper.

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

## Acknowledgements

This work was supported by the National Institute of Allergy and Infectious Diseases (NIAID)/National Institutes of Health (NIH) grants AI064798 and AI155674 (to M.J.) and by the Spanish Ministry of Science and Innovation grants BFU2016-77427-C2-2-R, PID2019-105649RB-I00 and PID2022-138694OB-I00 (to M.E.G.). The content is solely the responsibility of the authors and does not necessarily represent the official views of the NIH. We thank the Analytical Resources Core Facility at CSU (RRID: SCR_021758) for its help with LC/MS analyses, and Dr. Hataichanok (Mam) Scherman from the Protein Expression and Purification facility at CSU for the purification of PlsMsmg and PlsB2smg.

## Author contributions

S.K.A., D.K., C.H., A.Q., M.E.G. and M.J. designed research. S.K.A., A.C.G., E.H.C., D.K., I.A., V.J., Z.P., J.M.B., C.d.S., L.S. and N.S. performed research. S.K.A., E.H.C., D.K., V.J., M.M., M.E.G. and M.J. analyzed data. M.E.G. and M.J. wrote the main manuscript text. All authors reviewed the final version of the manuscript.

## Competing interests

The authors declare no competing interests.
