## [Peer review file · Nature Communications]

REVIEWER COMMENTS

Reviewer #1 (Remarks to the Author):

Review comments for Angala et al. Nature Communications

Angala and co-workers represented a unique pathway of glycerolipid acylation in Mycobacterium, through the identification and characterization of a mycobacterial glycerol-3-phosphate / 1-acyl-glycerol-3-phosphate sn-2 acyltransferase in Mycobacterium smegmatis. The authors identified potential phospholipid acyltransferase homologues in Mycobacterium using sequence similarity. They demonstrated the M. smegmatis gene, MSMEG_4248, which they renamed as PlsMsmg, is essential and further characterized gene and enzyme it encodes using conditional mutant, cell-free assays and molecular simulation.

The activity of an enzyme for sn-2 acylation of G3P renewed our biochemical knowledge on mycobacterial lipid metabolism, and unveiled a non-classical pathway for formation of phospholipids through sn-2 acylation of G3P, followed by sn-1 acylation. As an essential gene, detailed characterization of PlsM properties will also be relevant to future works including drug discovery. Hence the manuscript will be of interest to the broader community. However, several points need to be addressed before the works can be accepted for publication.

Comments:

1) The actual effects of PlsM on M. smegmatis was not very clear for a few reasons:

a) The authors had created a conditional mutant to characterize the effects of PlsM on lipid metabolism in M. smegmatis, which are subjected to tetracycline treatment. Can the authors clarify if the lipid effects they observed (figure 1 and associated supp. Data) are not due to 1) tetracycline treatment, and 2) stage of growth (in consideration the mutants have growth defects).

b) Related to the above point, there is a lack of details how the bacterium was cultured and sampled for lipid analysis. These factors are important and can affect the lipid composition. Moreover, there is no data shown for replication and in fact no indication of the biological replicate numbers.

c) It was described that PlsM play an essential role in early stages of glycerolipid synthesis leading to PA. However, it is not clear from the data how this is the case. Since PlsM is postulated to be important for PA formation, how do the levels of LPA (sn-1, sn-2) and PA as well as the fatty acid composition differ between wild type vs mutant. The methodologies (LCMS in particular) used in this work should be able to address this question.

d) From the TLC data (figure 1), it is clear that the major lipid classes, CL and PE were strongly downregulated (based on equal loading stated in Materials and Methods). However, the authors described a 50% decrease in TAG and a 35% increase in upon plsM silencing. Here the results are contradicting. However, this is attributed to the fact that the LCMS data which was used to describe the TAG and CL changes were normalised as relative abundance. This only tells the relative distribution of lipid classes (out of total signals) but do not tell absolute levels. Technically to determine the effects on phospholipid levels, the lipids should be (semi-)quantified and with equal loading (as per the TLC) to better reflect the effects of PlsM silencing on lipid levels, particularly phospholipids and glycerolipids to demonstrate how PlsM affects glycerolipids in mycobacterium. Indeed the effects of PlsM on Mycobacterial glycerolipids (including TAG, DAG and phospholipids) composition can be better clarified with LC-MS to derive a more coherent view.

e) From the cell based assays, it was evident that the substrate specificity for PlsM is C16:0. How about the actual amounts of C16:0 FA containing lipids in the conditional mutant?

2) The effects of PlsM replacement by PlsC was further studied. Similar to the above comments, the conditions for genetic manipulation can potentially affect membrane lipids. Hence, it will be clearer if proper controls are shown.

3) Related to the enzymatic activity of PlsM (Figure 3), the activity for G3P as a substrate is fairly low (15% conversion), in contrast to sn-1 LPA (>90%). In addition in Figure 3B, it seemed that sn-1 LPA was formed in addition to sn-2 LPA. This was most prominent when C18:1 CoA was used as donor

(albeit product formation is low but main product is sn-1 LPA). Can the authors address: (i) if PlsM has dual substrate specificity, and (ii) if sn-2 acylation of G3P can be mediated by other PlsC?

Gene silencing led to changes in phospholipids (including PIM) (decreased level, but increase TMM?)

Other comments:

Line 73, to clarify 'two first families'

Materials and method, to include

- clarifications on culture conditions, and how lipid amounts are determined for equal loading.
- How data is analysed for LCMS (normalization, quantitation etc)
- Biological replicates will be needed for bioanalysis of lipids

Generally with regards to LCMS analysis (which does form an important aspect of this study), the lipidomics reporting guideline (Introducing the Lipidomics Minimal Reporting Checklist | Nature Metabolism), as part of the community effort for traceability and reproducibility.

Discussion: How did the authors conclude PlsM is essential for the biosynthesis of all major forms of mycobacterial GPL?

Reviewer #2 (Remarks to the Author):

The manuscript (NSCOMMS-23-09379-T) untitled "A Unique Pathway to the Acylation of Glycerolipids in Mycobacteria" by Shiva Kumar Angala et al., describes the role of 2 unique acyltransferases PlsM and PlsB2 in mycobacteria involved in both acylation pathway of the sn-1 (sn-2) position of Glycero-3-phosphate and then the remaining sn-2 (sn-1) position. The paper is very well written, experiments are very convincing and provide many information on the physiological role of these enzymes. The mass spectrometry analyses allow to decipher the lipid and fatty acids transfer by both enzymes and the complexity of these acyltransferases. Knowing that glycerol-3-phosphate is as a central molecule involved in numerous processes, including glycerophospholipid and triacylglycerol metabolism, have the authors considered to complete the paper by studying the intracellular TAG synthesis i.e. intracellular lipid inclusions which are known to play an important role in the physiology of mycobacteria?

Except this latter point, I do not have any major concerns about this work that could be accepted in Nature Communications. Moreover, to improve the quality of the paper, some minor corrections have to be made.

1-Authors claim they performed all experiments in triplicate. Please indicate Standard deviation and give some statistical analysis when it is needed (Figure 2, S1, S4, S6).

2-Biochemical data of PlsC from *E. coli* and PlsC from *B. subtilis* (% of identity between both PlsC and substrat specificity) obtained from the literature could be added in the text to better understand the choice of both enzymes which is not only based on the % of sequence Identity.

Authors suggest that "the disruption of the plsM locus of Msmg was achievable in the presence of the plsC gene from *E. coli* (plsCcoli) but not that from *B. subtilis* (plsCsubtilis) [Fig. 2A] despite comparable levels of expression of plsCcoli and plsCsubtilis in *M. smegmatis* [Fig. S2]", the authors have to check at least the presence of the protein by western blot.

2-Where is the information in the table S1 "In light of the fact PlsM presents the characteristics of a 1-acyl-G3P acyltransferase (PlsC-type enzyme) and, thus, of an enzyme transferring acyl chains specifically to position sn-2 [Table S1]", please clarify.

3-Figure 1: this figure could be modified by adding Figure S1, reformat Table S2 as a graph inserted directly in Figure 1. Moreover, the C panel could be moved to supplemental material.

4-B panel in Figure 2: the mutant and WT curve growth being totally different (Figure S3), did lipid extraction performed at a given time or equivalent OD to be at the same physiological state of growth? This must be clearly precise in the material and method section.

5-Figure 3: for a better understanding, the metabolic pathway presented in (e) in 3B should be added in a single panel (Figure 3A) to depict the chemical reaction with PlsB and PlsM.

6-Line 222-224: About the ability to transfer C18:1 and C18:0, do you think that 0.4 ± 0.3 % yield is

really relevant (Figure 3A and 4B)? Many argues should be added to be convincing.

7-Table S3, except for the MIC value for Ciprofloxacin which are comparable could you explain the MIC values of the *M. smegmatis* WT in this table which are very high compared to literature data (see Li et al. doi: 10.1128/AAC.48.7.2415-2423.2004) where they found a MIC of 1 µg/mL for the RIF, 8 for the INH and 8 for the CHL versus 80, 32-16 and 31-62 µg/mL respectively.

8-The paragraph in the discussion section lines 317-323, should be moved to the result section to support experimental specificity data.

Reviewer #3 (Remarks to the Author):

The manuscript by Shiva Kumar Angala and co-workers aims to unravel the unique biosynthetic pathways of phosphatidic acids in mycobacteria. Indeed, mycobacteria appear to have developed unique ways of synthesising phosphatidic acids that differ from other bacteria in many ways, in particular by using acyl-CoA rather than acyl-ACP as donor substrates and by showing a specific distribution of fatty acids at sn1 and sn2 positions. The manuscript provides novel and very important clues to the understanding of this unique pathway by identifying two acyltransferases that appear to have exquisite enzymatic specificities for both the donor substrate and the hydroxyl positions of the acceptor substrates. To achieve this, the authors have elegantly combined silencing and rescuing experiments on both genes in *Mycobacterium smegmatis*, detailed structural analysis of the lipid profiles of mutant strains and biochemical analysis of the recombinantly expressed enzymes. Overall, this is a straightforward and well written manuscript with data from high quality experiments that appear technically sound and most of the claims are supported by the data, despite the difficulties associated with the known migration of acyl groups on sn1 and sn2 glycerophospholipids. This reviewer believes that this work will potentially advance knowledge of mycobacterial lipid biosynthesis. The following comments could be considered to clarify some of the experimental approaches and conclusions.

1- The results of the lipid profiling of *Msmeg*-Delta-plsM strain grown under permissive and non-permissive conditions obtained by TLC (Fig. 1D) and LC/MS (Fig. S1A) appear to be somewhat antagonistic. This is particularly the case for CL, which seems to follow different trends. Would the authors be so kind as to explain this apparent discrepancy?

2- Considering that the *Msmeg*-Delta-plsM strain was shown to have a longer lag period than the WT strain (Fig. S4A) and to have a different lipid content at steady state, it may be useful to compare the lipid content along the exponential phases of both strains to gain a better understanding of the dynamics of the lipid shift.

3- The authors convincingly demonstrated that PlsM was able to transfer acyl chain on sn-2 position of both G3P and Sn-1-LPA substrates, as shown in Fig 3. However, they failed to clearly establish that it could not transfer acyl chain at the sn-1 position and mostly assumed this to be the case based on sequence homologies (Table S1). Their hypothesis would be greatly strengthened if they could unambiguously demonstrate this, for example by using a similar experimental approach for PlsB2 as shown in Figure 4.

4- Sugasini and Subbaiah (ref 21) show that acyl migration in LPC is strongly dependent on the nature of the acyl groups, pH and temperature. In that report, the migration appears to be much slower than that observed in the present manuscript under standard conditions, as shown in Figures 3Ba and 4A. Could the authors explain this apparent discrepancy in the results? Could they also comment on the possible in vivo relevance of this migration in bacteria?

5- Figure 4A, which describes the generation of a mixture of sn-1 and sn-2 LPA upon digestion with phospholipase A1, should be mentioned earlier as it is also relevant to the understanding of Figure 3. This would make the manuscript easier to read.

6- The overall modification of glycerolipids synthesis (decrease and increased incorporations) observed in cell free assays (Figure 5) would certainly gain to be better explained.

7- The modelling experiments that support the mechanisms of substrate recognition should be at least partially included in the results section, as this is a very important piece of information that strengthens the authors' conclusions.

Response to Reviewers

Reviewer #1

Angala and co-workers represented a unique pathway of glycerolipid acylation in Mycobacterium, through the identification and characterization of a mycobacterial glycerol-3-phosphate / 1-acyl-glycerol-3-phosphate sn-2 acyltransferase in Mycobacterium smegmatis. The authors identified potential phospholipid acyltransferase homologues in Mycobacterium using sequence similarity. They demonstrated the M. smegmatis gene, MSMEG_4248, which they renamed as PlsMsmg, is essential and further characterized gene and enzyme it encodes using conditional mutant, cell-free assays and molecular simulation.

The activity of an enzyme for sn-2 acylation of G3P renewed our biochemical knowledge on mycobacterial lipid metabolism, and unveiled a non-classical pathway for formation of phospholipids through sn-2 acylation of G3P, followed by sn-1 acylation. As an essential gene, detailed characterization of PlsM properties will also be relevant to future works including drug discovery. Hence the manuscript will be of interest to the broader community. However, several points need to be addressed before the works can be accepted for publication.

We thank the reviewer for her/his positive assessment of our work and constructive comments.

Comments:

- 1) The actual effects of PlsM on M. smegmatis was not very clear for a few reasons:
 - a) The authors had created a conditional mutant to characterize the effects of PlsM on lipid metabolism in M. smegmatis, which are subjected to tetracycline treatment. Can the authors clarify if the lipid effects they observed (figure 1 and associated supp. Data) are not due to 1) tetracycline treatment, and 2) stage of growth (in consideration the mutants have growth defects.
 - b) Related to the above point, there is a lack of details how the bacterium was cultured and sampled for lipid analysis. These factors are important and can affect the lipid composition. Moreover, there is no data shown for replication and in fact no indication of the biological replicate numbers.

The lipid analyses conducted on the M. smegmatis control strain and plsM conditional mutants were purposely performed in multiple ways to better assess consistency/trends across different culture conditions and anhydro-tetracycline (ATc) concentrations. Both liquid and agar cultures were used because, based on our experience, gene silencing in mycobacteria is often more efficient and stable over time on solid medium. Details were added to the Figure 1 legend and in the results section to describe which specific strains were used, how the bacteria were grown and when they were collected for lipid analyses. The number of repeat experiments are now also indicated in the figure legends.

Across all experiments shown in Fig. 1 and Table S2 (TLC, LC/MS and GC/MS analyses), the conditional plsM knock-downs grown in the presence of different concentrations of ATc were compared to similarly treated control cultures (Msmg/pSETetR) to distinguish potential effects

of ATc treatment from that of *plsM* silencing on the observed glycerolipid phenotypes. ATc by itself was found to have no significant impact on the lipid content of the control strain.

The replication rate of the control and conditional knock-down grown in 7H9-OADC-tyloxapol in the presence of different concentrations of ATc (triplicate cultures) is now shown in Fig. 1D.

The TLC analyses shown in Fig. 1 were conducted on bacteria grown on agar plates and collected on the same day (as shown in Fig. 1C). The new quantitative LC/MS analysis shown in Fig. 1F-G (and GC/MS analysis shown in Table S2) was conducted on bacteria grown on liquid medium (duplicate cultures) and collected at the same OD (~ 0.5 to 0.6) as detailed in the figure legend.

Overall, our results show that (i) the specific decrease in glycerolipids in the conditional knock-down upon *plsM* silencing is reproducible across experiments, and (ii) ATc concentration-dependent.

c) It was described that PlsM play an essential role in early stages of glycerolipid synthesis leading to PA. However, it is not clear from the data how this is the case. Since PlsM is postulated to be important for PA formation, how do the levels of LPA (sn-1, sn-2) and PA as well as the fatty acid composition differ between wild type vs mutant. The methodologies (LCMS in particular) used in this work should be able to address this question.

LC/MS-based data for the PA content of the *plsM* knockdown mutant and control strains grown in the presence of different concentrations of ATc are presented in Fig. 1F (inset). PA was present in very low quantities in the samples and the content did not significantly differ between strains or between growth conditions (i.e., ATc concentrations). LPA was not detected in any of the samples. The stated conclusion is based on the decrease of all forms of glycerophospholipids upon *plsM* silencing which points to the common PA precursor production being affected.

d) From the TLC data (figure 1), it is clear that the major lipid classes, CL and PE were strongly downregulated (based on equal loading stated in Materials and Methods). However, the authors described a 50% decrease in TAG and a 35% increase in upon *plsM* silencing. Here the results are contradicting. However, this is attributed to the fact that the LCMS data which was used to describe the TAG and CL changes were normalised as relative abundance. This only tells the relative distribution of lipid classes (out of total signals) but do not tell absolute levels. Technically to determine the effects on phospholipid levels, the lipids should be (semi-)quantified and with equal loading (as per the TLC) to better reflect the effects of PlsM silencing on lipid levels, particularly phospholipids and glycerolipids to demonstrate how PlsM affects glycerolipids in mycobacterium. Indeed the effects of PlsM on Mycobacterial glycerolipids (including TAG, DAG and phospholipids) composition can be better clarified with LC-MS to derive a more coherent view.

The reviewer is correct in her/his interpretation of the apparent discrepancy between the TLC and LC/MS results. The previous version of Fig. S1 indeed presented relative abundances and, thus, tended to show that CL was relatively less affected than other glycerolipid forms by *plsM*

silencing (standardization was to the total glycerolipid signals). To better resolve the effects of *plsM* silencing on glycerolipid levels, a repeat quantitative LC/MS experiment performed in biological duplicate is now presented in Fig. 1F that shows absolute levels of glycerolipids under permissive and non-permissive growth conditions.

e) From the cell based assays, it was evident that the substrate specificity for PlsM is C16:0. How about the actual amounts of C16:0 FA containing lipids in the conditional mutant?

The GC/MS analysis of fatty acid methyl esters derived from total lipids prepared from the control and conditional knockdown is presented in Table S2. The results revealed a ~ 20% decrease in palmitic acid content when *plsM* is silenced.

LC/MS-based analysis of free fatty acids in the same strains (see Fig. 1F; inset) failed to reveal any accumulation of C16:0 (or other free fatty acids) in the cells upon *plsM* silencing when compared to the control strain.

2) The effects of PlsM replacement by PlsC was further studied. Similar to the above comments, the conditions for genetic manipulation can potentially affect membrane lipids. Hence, it will be clearer if proper controls are shown.

The reviewer's point is well taken. We actually tried to engineer a control strain with a closer genetic background to *MsmgΔplsM/pMVG1-plsCcoli* by generating a *MsmgΔplsM* strain rescued with *plsMtb* expressed from the same replicative (multicopy) pMVG1 plasmid and expressed from the strong, constitutive, mycobacterial promoter *Phsp60*. The expression of *plsMtb* from this plasmid, however, turned out to be toxic and we were thus not able to generate the sought control.

To mitigate this potential issue, we are now comparing side-by-side in the metabolic labeling experiment shown in Fig. 2C *Msmg* WT harboring an empty pMVG1 plasmid to *MsmgΔplsM/pMVG1-plsCcoli* (two independent clones, # 32 and 34) and *MsmgΔplsM/pSETetR-plsMtb* (grown in the presence of 50 ng/mL ATc to induce *plsMtb* expression). The results show that both control strains, *Msmg/pMVG1* and *MsmgΔplsM/pSETetR-plsMtb* (50 ng/mL ATc), despite their different genetic backgrounds, display similar (WT) phospholipid biosynthetic profiles and rates that significantly differ from those of the two *plsCcoli*-rescued *plsM* mutant clones (# 32 and 34). The growth curves for the same four strains is presented in Fig. 2B.

3) Related to the enzymatic activity of PlsM (Figure 3), the activity for G3P as a substrate is fairly low (15% conversion), in contrast to sn-1 LPA (>90%). In addition in Figure 3B, it seemed that sn-1 LPA was formed in addition to sn-2 LPA. This was most prominent when C18:1 CoA was used as donor (albeit product formation is low but main product is sn-1 LPA). Can the authors address: (i) if PlsM has dual substrate specificity, and (ii) if sn-2 acylation of G3P can be mediated by other PlsC?

It indeed appears like the less efficient the acyl donor (i.e. C18:1-CoA is less efficiently used than C18:0-CoA and C16:0-CoA based on the data shown on Fig. 3C), the lower the sn-2 to sn-1-LPA

product ratio. We believe the small quantity of *sn*-1 LPA products formed to result from the spontaneous transmigration of the acyl chain enzymatically transferred to position *sn*-2 by PlsM (which, unfortunately, we are not able to block completely). This assumption is based on the apparent preferred transfer of C16:0 to position *sn*-2 of G3P by PlsM (Fig. 3C), and the results of new assays now presented in Fig. 4 (in response to one of Reviewer 3's comments) showing that PlsM is apparently not able to transfer acyl chains to position *sn*-1 of *sn*-2-LPA. Thus, we believe that a dual positional specificity of PlsM is unlikely. A comment to this effect was added to the text on p. 7, lines 215-217.

Of note, the spontaneous transmigration rate of acyl chains has been shown to vary not only with the type of acceptor substrate (head group of lysophospholipids in particular), but also with the chain length and degree of unsaturation of the acyl chain. Although there is, to the best of our knowledge, no published data comparing the transmigration rate of C18:1 to that of C18:0 or C16:0 on LPA, it is possible that C18:1 within *sn*-2 C18:1-LPA is more prone to spontaneous transmigration.

Based on the results presented in Fig. 2E (and p. 7, lines 213-217), it is likely that PlsM is the sole mycobacterial PlsC homolog responsible for the acylation of position *sn*-2 of G3P. However, due to spontaneous acyl transmigration that we cannot totally block, it is not possible to completely exclude that the low amount of *sn*-2 LPA product detected in membrane preparations from *MsmgΔplsM/pMVGH1-plsCcoli* may result from residual *sn*-2 acyltransferase activity coming from other PlsCs.

Gene silencing led to changes in phospholipids (including PIM) (decreased level, but increase TMM?)

Since approximately the same total amount of lipids was loaded per lane on the TLC shown in Fig. 1E, an apparent relative increase in TMM is observed when phospholipid levels decrease. For quantitative LC/MS-based analyses of PIM contents, please see the revised Fig. 1 (panel F).

Other comments:

Line 73, to clarify 'two first families'

Addressed. p. 3, line 75-76.

Materials and method, to include

- clarifications on culture conditions, and how lipid amounts are determined for equal loading.
- Biological replicates will be needed for bioanalysis of lipids.

Details were added to the main text and figure legends to clarify culture conditions, lipid analyses and biological repeats/reproducibility. Fig. 1 and former Fig. S1 were significantly revised as detailed above.

Generally with regards to LCMS analysis (which does form an important aspect of this study), the lipidomics reporting guideline (Introducing the Lipidomics Minimal Reporting Checklist | Nature Metabolism), as part of the community effort for traceability and reproducibility.

We followed the suggested guidelines and the output checklist is now provided as a supplementary file (Supplementary File S1) in the Supplementary Materials.

Discussion: How did the authors conclude PlsM is essential for the biosynthesis of all major forms of mycobacterial GPL?

This conclusion is based on the decrease of all glycerolipid forms that follows *plsM* silencing as now shown in the revised Fig. 1 (see response to point 1.c) and the demonstrated enzyme activity of PlsM. The function of PlsM is consistent with its essentiality for mycobacterial growth.

Reviewer #2:

The manuscript (NSCOMMS-23-09379-T) untitled “A Unique Pathway to the Acylation of Glycerolipids in Mycobacteria” by Shiva Kumar Angala et al., describes the role of 2 unique acyltransferases PlsM and PlsB2 in mycobacteria involved in both acylation pathway of the sn-1 (sn-2) position of Glycerol-3-phosphate and then the remaining sn-2 (sn-1) position. The paper is very well written, experiments are very convincing and provide many information on the physiological role of these enzymes. The mass spectrometry analyses allow to decipher the lipid and fatty acids transfer by both enzymes and the complexity of these acyltransferases. Knowing that glycerol-3-phosphate is as a central molecule involved in numerous processes, including glycerophospholipid and triacylglycerol metabolism, have the authors considered to complete the paper by studying the intracellular TAG synthesis i.e. intracellular lipid inclusions which are known to play an important role in the physiology of mycobacteria?

We thank the reviewer for his/her positive assessment of our work. The impact of *plsM* silencing on TAG content under standard laboratory growth conditions is shown in Fig. 1. As expected from the function of PlsM, TAG synthesis is as dramatically affected upon *plsM* silencing as that of other glycerolipids.

TAG-containing intracellular lipid inclusions are typically formed under conditions of non-replicating persistence. We haven't tried to silence *plsM* under these conditions. Analyzing the TAG content of the *plsM* cKD under non-replicating persistence would certainly be something of interest in the context of future physiological studies. We thank the reviewer for this suggestion.

Except this latter point, I do not have any major concerns about this work that could be accepted in Nature Communications. Moreover, to improve the quality of the paper, some minor corrections have to be made.

1-Authors claim they performed all experiments in triplicate. Please indicate Standard deviation and give some statistical analysis when it is needed (Figure 2, S1, S4, S6).

The specifics of how many biological and technical replicates are now included in the legends accompanying each of the figures and supplementary figures.

Some experiments were repeated and used to revise Fig. 1 and Fig. 2 as suggested by reviewer 1. In some cases (e.g., *MsmgΔplsM* rescued with *plsCcoli*), we opted to analyze two

independent recombinant clones rather than or in addition to repeating experiments on the same clone to emphasize reproducibility. We also favored biological replicates over technical replicates. Because of the slight variability that exists from batch to batch and clone to clone in terms of absolute lipid or fatty acid levels, it is not possible to add error bars and perform statistical analyses on some of the graphs (e.g., Fig. 2D).

2-Biochemical data of PlsC from *E. coli* and PlsC from *B. subtilis* (% of identity between both PlsC and substrat specificity) obtained from the literature could be added in the text to better understand the choice of both enzymes which is not only based on the % of sequence Identity.

We have amended the results section (p. 5, lines 148-150) to include three original references describing the biochemical characteristics of PlsC from *E. coli* and *B. subtilis*.

Authors suggest that “the disruption of the *plsM* locus of *Msmg* was achievable in the presence of the *plsC* gene from *E. coli* (*plsCcoli*) but not that from *B. subtilis* (*plsCsubtilis*) [Fig. 2A] despite comparable levels of expression of *plsCcoli* and *plsCsubtilis* in *M. smegmatis* [Fig. S2]”, the authors have to check at least the presence of the protein by western blot.

Unfortunately, we tried but failed to detect recombinant *PlsCcoli* and *PlsCsubtilis* production in *Msmg* by immunoblot using anti-hexahistidine tag antibodies, reason why we opted for RT-qPCR instead. We believe this to be due to the low level of expression of the two heterologous genes.

2-Where is the information in the table S1 “In light of the fact *PlsM* presents the characteristics of a 1-acyl-G3P acyltransferase (*PlsC*-type enzyme) and, thus, of an enzyme transferring acyl chains specifically to position sn-2 [Table S1]”, please clarify.

Details were added to Table S1 and in the text (p. 4, lines 100-101) to better explain how the mycobacterial *PlsB* and *PlsC* candidates were identified based on Enzyme Commission and Gene Ontology numbers in addition to primary sequence similarity.

3-Figure 1: this figure could be modified by adding Figure S1, reformat Table S2 as a graph inserted directly in Figure 1. Moreover, the C panel could be moved to supplemental material.

Figure 1 was reorganized within the constraints of space limitation. Panel C was retained since it clearly illustrates the growth arrest that follows *plsM* silencing and shows the CFUs whose lipid content was analyzed by TLC in panel E. A panel D was added showing growth curves in liquid medium as requested by reviewer # 1. Quantitative LC/MS data showing the lipid changes that occur in the cKD following *plsM* silencing were added as panel 1F. Due to space limitation, the fatty acid methyl ester data remain in Table S2 but changes in the degree of unsaturation of glycerophospholipids is now presented in Fig. 1G.

4-B panel in Figure 2: the mutant and WT curve growth being totally different (Figure S3), did lipid extraction performed at a given time or equivalent OD to be at the same physiological state of growth? This must be clearly precise in the material and method section.

Fig. 2B (now Fig. 2C) shows the result of a metabolic labeling experiment with ^{14}C -acetate performed over the course of 8 hours. As now described in the figure legend, ^{14}C -acetate was added to cultures grown to an OD600 nm of 0.8, at which point the two control strains and two *MsmgΔplsM* clones rescued with *plsCcoli* display the same replication rate as now shown in Fig. 2B.

5-Figure 3: for a better understanding, the metabolic pathway presented in (e) in 3B should be added in a single panel (Figure 3A) to depict the chemical reaction with PlsB and PlsM.

This figure was reorganized as suggested.

6-Line 222-224: About the ability to transfer C18:1 and C18:0, do you think that 0.4 ± 0.3 % yield is really relevant (Figure 3A and 4B)?? Many argues should be added to be convincing.

We agree with the reviewer that the measured activity is very low activity and should probably not be considered significant based solely on Fig. 3. The problem is that this assay is not quantitative since the *sn-2* LPA substrate used by the PlsB2 enzyme results from the spontaneous transmigration of C16:0 from position *sn-1* to position *sn-2* which we do not have any control on. Fig. 5 solely provides qualitative evidence that C18:0 can be transferred by PlsB2 to position *sn-1* of *sn2*-LPA. The sentence on p. 8 (line 236) was modified to suggest that C18:1 is in fact a more efficient acyl donor than C18:0 for PlsB2 in these reactions.

7-Table S3, except for the MIC value for Ciprofloxacin which are comparable could you explain the MIC values of the *M. smegmatis* WT in this table which are very high compared to literature data (see Li et al. doi: 10.1128/AAC.48.7.2415-2423.2004) where they found a MIC of 1 $\mu\text{g}/\text{mL}$ for the RIF, 8 for the INH and 8 for the CHL versus 80, 32-16 and 31-62 $\mu\text{g}/\text{mL}$ respectively.

MIC values can vary with the medium used and readout (resazurin blue vs culture OD600 nm). We repeated this experiment with both clones of *MsmgΔplsM/pMVGH1-plsM* (clones 32 and 34) using a slightly different medium (7H9-ADC-tyloxapol) devoid of oleic acid and the results are presented in the revised Table S3. Slightly lower MICs were obtained for INH and STR. RIF MICs remain on the high side but are consistent with what we typically find with the *M. smegmatis* mc²155 strain used in our lab (e.g., <https://pubmed.ncbi.nlm.nih.gov/31110045/>).

8-The paragraph in the discussion section lines 317-323, should be moved to the result section to support experimental specificity data.

The section relative to the structural modeling of PlsM and PlsB2 was moved to the results section as suggested. The figures were revised accordingly (see new Fig. 7 and Fig. S8).

Reviewer #3:

The manuscript by Shiva Kumar Angala and co-workers aims to unravel the unique biosynthetic

pathways of phosphatidic acids in mycobacteria. Indeed, mycobacteria appear to have developed unique ways of synthesising phosphatidic acids that differ from other bacteria in many ways, in particular by using acyl-CoA rather than acyl-ACP as donor substrates and by showing a specific distribution of fatty acids at sn1 and sn2 positions. The manuscript provides novel and very important clues to the understanding of this unique pathway by identifying two acyltransferases that appear to have exquisite enzymatic specificities for both the donor substrate and the hydroxyl positions of the acceptor substrates. To achieve this, the authors have elegantly combined silencing and rescuing experiments on both genes in *Mycobacterium smegmatis*, detailed structural analysis of the lipid profiles of mutant strains and biochemical analysis of the recombinantly expressed enzymes. Overall, this is a straightforward and well written manuscript with data from high quality experiments that appear technically sound and most of the claims are supported by the data, despite the difficulties associated with the known migration of acyl groups on sn1 and sn2 glycerophospholipids. This reviewer believes that this work will potentially advance knowledge of mycobacterial lipid biosynthesis. The following comments could be considered to clarify some of the experimental approaches and conclusions.

1- The results of the lipid profiling of Msmeg-Delta-plsM strain grown under permissive and non-permissive conditions obtained by TLC (Fig. 1D) and LC/MS (Fig. S1A) appear to be somewhat antagonistic. This is particularly the case for CL, which seems to follow different trends. Would the authors be so kind as to explain this apparent discrepancy?

This apparent discrepancy is due to the fact that the original Fig. S1 was presenting relative abundances of glycerolipids in the control strain and conditional knock-down mutant grown under permissive and non-permissive conditions rather than absolute quantities. In other words, Fig. S1 was essentially showing that some forms of glycerolipids were more dramatically affected by the silencing of *plsM* than others. Fig. 1 and Fig. S1 have been revised to address this issue and provide more quantitative information as requested by reviewer # 1 (see new Fig. 1F and 1G).

2- Considering that the Msmeg-Delta-plsM strain was shown to have a longer lag period than the WT strain (Fig. S4A) and to have a different lipid content at steady state, it may be useful to compare the lipid content along the exponential phases of both strains to gain a better understanding of the dynamics of the lipid shift.

Fig. S4 (now Fig. S2) shows the lipid profiles of both strains during log phase (for cells collected at OD ~ 0.8) as now indicated in the figure legend. Rather than comparing the lipid content of both strains along the exponential phase of growth, the dynamics of lipid synthesis was addressed by the metabolic labeling experiment presented in Fig. 2C comparing *de novo* lipid synthesis in the two strains (plus a new control strain with a closer genetic background to *MsmgΔplsM/pMVGH1-plsCcoli* as suggested by reviewer # 1) during the exponential phase of growth over a period of 8 hours. The [¹⁴C-acetate] radiotracer was added to cultures that had grown to OD ~ 0.8 so metabolic incorporation could be monitored at a stage of the growth curves when the control and test strains display similar replication rates (see Fig. 2B).

Even though performing (non-radiolabeled) lipid analyses at different stages of the exponential phase may reveal more pronounced differences between strains for some lipid forms at certain time points, we believe they will not change the main conclusions of this experiment which are that the strain rescued with *plsCcoli* synthesizes all forms of glycerolipids found in the control strains expressing *plsM*, albeit at a reduced rate and generally containing less C16:0 and tuberculostearic acid and more of the unsaturated C18:1.

3- The authors convincingly demonstrated that PlsM was able to transfer acyl chain on sn-2 position of both G3P and Sn-1-LPA substrates, as shown in Fig 3. However, they failed to clearly establish that it could not transfer acyl chain at the sn-1 position and mostly assumed this to be the case based on sequence homologies (Table S1). Their hypothesis would be greatly strengthened if they could unambiguously demonstrate this, for example by using a similar experimental approach for PlsB2 as shown in Figure 4.

Thank you for this suggestion. The experiment suggested by the reviewer was performed and is now shown as Fig. 4. The results, which are now discussed on p. 7, lines 213-217 of the Results section, support the conclusion that PlsM is not able to transfer an acyl chain to position sn-1 of 1-hydroxy-2-palmitoyl-sn-G3P, contrary to PlsB2.

4- Sugasini and Subbaiah (ref 21) show that acyl migration in LPC is strongly dependent on the nature of the acyl groups, pH and temperature. In that report, the migration appears to be much slower than that observed in the present manuscript under standard conditions, as shown in Figures 3Ba and 4A. Could the authors explain this apparent discrepancy in the results? Could they also comment on the possible in vivo relevance of this migration in bacteria?

The conditions used in our study to mitigate spontaneous acyl chain transmigration follow the procedure recommended by Sugasini and Subbaiah and we therefore do not have any obvious explanation for the more rapid transmigration of the acyl chains observed in the case of our LPA products. Unfortunately, to the best of our knowledge, there has been no published study looking at the rate of spontaneous acyl transmigration in LPA. Published studies thus far have analyzed acyl transmigration either in monoglycerides (but with saturated acyl chain no longer than C16:0, Boswinkel *et al.*, 1996) or lysophospholipids (Kawana *et al.*, 2014; Okudaira *et al.*, 2014; Sugasini *et al.*, 2017). Since these studies have shown that the spontaneous transmigration rate of acyl chains varied not only with the chain length and degree of unsaturation of the acyl chain but also the head group of lysophospholipids, one may hypothesize that the transmigration rate is greater in LPA than in lysophospholipids. This assumption is supported by the apparent lack of finding of sn-2-LPA products in the biological samples analyzed by Okudaira *et al.* (see Table 5; only sn-1-LPAs were found), while both sn-1 and sn-2 lysophospholipids were detected.

Whether spontaneous acyl transmigration occurs in live bacteria is hard to tell. One would expect the balance of acyltransferase and phospholipase A1/A2 activities in the cells to maintain

the pool of *sn*-1 LPA/*sn*-2-LPAs and positional distribution of fatty acids in glycerolipids at optimal levels for growth and survival.

- Boswinkel, G., Derksen, J. T. P., vantRiet, K., and Cuperus, F. P. (1996) Kinetics of acyl migration in monoglycerides and dependence on acyl chainlength. *J Am Oil Chem Soc* **73**, 707-711
- Kawana, H., Kano, K., Shindou, H., Inoue, A., Shimizu, T., and Aoki, J. (2019) An accurate and versatile method for determining the acyl group-introducing position of lysophospholipid acyltransferases. *Biochim Biophys Acta Mol Cell Biol Lipids* **1864**, 1053-1060
- Okudaira, M., Inoue, A., Shuto, A., Nakanaga, K., Kano, K., Makide, K., Saigusa, D., Tomioka, Y., and Aoki, J. (2014) Separation and quantification of 2-acyl-1-lysophospholipids and 1-acyl-2-lysophospholipids in biological samples by LC-MS/MS. *J Lipid Res* **55**, 2178-2192
- Sugasini, D., and Subbaiah, P. V. (2017) Rate of acyl migration in lysophosphatidylcholine (LPC) is dependent upon the nature of the acyl group. Greater stability of *sn*-2 docosahexaenoyl LPC compared to the more saturated LPC species. *PLoS One* **12**, e0187826

5- Figure 4A, which describes the generation of a mixture of *sn*-1 and *sn*-2 LPA upon digestion with phospholipase A1, should be mentioned earlier as it is also relevant to the understanding of Figure 3. This would make the manuscript easier to read.

We believe that the addition of a new experiment in response to this reviewer's point # 3 (presented as Fig. 4 with comments in the results section on p. 7, lines 213-217) helps address this issue.

6- The overall modification of glycerolipids synthesis (decrease and increased incorporations) observed in cell free assays (Figure 5) would certainly gain to be better explained.

This section was rewritten to improve clarity (p. 8, lines 253-261).

7- The modelling experiments that support the mechanisms of substrate recognition should be at least partially included in the results section, as this is a very important piece of information that strengthens the authors' conclusions.

The section relative to the structural modeling of PlsM and PlsB2 was moved to the results section as suggested. The figures were revised accordingly (see new Fig. 7 and Fig. S8).

REVIEWERS' COMMENTS

Reviewer #1 (Remarks to the Author):

Review comments for Angala et al. Nature Communications (Revision)

Angala and co-workers represented a unique pathway of glycerolipid acylation in Mycobacterium, through the identification and characterization of two mycobacterial acyl transferases acyltransferase in Mycobacterium smegmatis: (i) PlsMmsg: glycerol-3-phosphate / 1-acyl-glycerol-3-phosphate sn-2, and (ii) PlsB2, which acts on the sn-1 position of G3P and sn-2 LPA. The authors identified the potential phospholipid acyltransferase homologues in Mycobacterium using sequence similarity, and further characterized the genes and enzymes it encodes using bacterial genetics, biochemistry, lipidomics, cell-free assays and molecular simulation.

In the current revision, the authors had addressed the concerns raised. They had now provided additional data which further substantiated the roles of PlsM and PlsB2 in mycobacterial glycerol(phospho)lipid remodelling, and confirmed the substrate specificity of these enzymes in vitro. The authors had also provided clearer indications on the number of replicates used in the study. The study involving the unravelling of the biochemistry of mycobacterial lipid remodelling, which is an essential process for the bacterium, is of general interest to a broad community, and will be ready for acceptance with the following comments addressed.

- 1) For the data we appreciate the clarification of the numbers of replicates. Appropriate statistical tests should be in place throughout the manuscript. E.g in Figure 1, panel F had error bars and significance tested, but not for panel G.
- 2) The data in figure 1 is convincing to conclude PlsM is involved in glycerol(phospho) lipid. For the statement 'Altogether, the data are thus consistent with plsM playing an essential role in the early stages of glycerolipid synthesis leading to PA.', it will be more suitable based on the later biochemical experiments which were more targeted at PA remodelling.
- 3) Figure 1D has no legend.
- 4) Figure 1G and S2B: degree of unsaturation should be for sum composition and be clearly labelled/indicated.
- 5) To clarify line 213-214. 'The rapid transmigration of acyl chains from position sn-2 to position sn-1 of LPA makes it to exclude that PlsM may transfer acyl chains to position sn-1 in addition to position sn-2 of G3P.'

Reviewer #2 (Remarks to the Author):

After reading the rebuttal and the full revised version of the manuscript, this work can be therefore accepted for publication.

From my point of view, authors answered clearly to all questions I have addressed and improved the manuscript as expected

Reviewer #3 (Remarks to the Author):

The authors have responded adequately to all my comments. The manuscript is of very high quality and scientific relevance and I recommend it for publication as it stands.

REVIEWERS' COMMENTS

We thank the reviewers for their time and constructive comments.

Reviewer #1 (Remarks to the Author):

Review comments for Angala et al. Nature Communications (Revision)

Angala and co-workers represented a unique pathway of glycerolipid acylation in Mycobacterium, through the identification and characterization of two mycobacterial acyl transferases acyltransferase in Mycobacterium smegmatis: (i) PlsMmsg: glycerol-3-phosphate / 1-acyl-glycerol-3-phosphate sn-2, and (ii) PlsB2, which acts on the sn-1 position of G3P and sn-2 LPA. The authors identified the potential phospholipid acyltransferase homologues in Mycobacterium using sequence similarity, and further characterized the genes and enzymes it encodes using bacterial genetics, biochemistry, lipidomics, cell-free assays and molecular simulation.

In the current revision, the authors had addressed the concerns raised. They had now provided additional data which further substantiated the roles of PlsM and PlsB2 in mycobacterial glycerol(phospho)lipid remodelling, and confirmed the substrate specificity of these enzymes in vitro. The authors had also provided clearer indications on the number of replicates used in the study. The study involving the unravelling of the biochemistry of mycobacterial lipid remodelling, which is an essential process for the bacterium, is of general interest to a broad community, and will be ready for acceptance with the following comments addressed.

1) For the data we appreciate the clarification of the numbers of replicates. Appropriate statistical tests should be in place throughout the manuscript. E.g in Figure 1, panel F had error bars and significance tested, but not for panel G.

Error bars were actually included in Figure 1G but are too small in most instances to clearly show. Statistical significance was added as requested.

2) The data in figure 1 is convincing to conclude PlsM is involved in glycerol(phospho) lipid. For the statement 'Altogether, the data are thus consistent with plsM playing an essential role in the early stages of glycerolipid synthesis leading to PA.', it will be more suitable based on the later biochemical experiments which were more targeted at PA remodelling.

We have amended the sentence as follows in the hope it appropriately addresses the reviewer's comment: 'Altogether, the data are thus consistent with plsM playing an essential role in the early stages of glycerolipid synthesis.'

3) Figure 1D has no legend.

The legend of figure 1D was fixed.

4) Figure 1G and S2B: degree of unsaturation should be for sum composition and be clearly labelled/indicated.

The legends of Figure 1G and S2B were edited to improve clarity.

5) To clarify line 213-214. 'The rapid transmigration of acyl chains from position sn-2 to position sn-1 of LPA makes it to exclude that PlsM may transfer acyl chains to position sn-1 in addition to position sn-2 of G3P.

We have amended the sentence as follows: 'The rapid transmigration of acyl chains from position sn-2 to position sn-1 of LPA makes it impossible to exclude that PlsM may transfer acyl chains to position sn-1 in addition to position sn-2 of G3P.

Reviewer #2 (Remarks to the Author):

After reading the rebuttal and the full revised version of the manuscript, this work can be therefore accepted for publication.

From my point of view, authors answered clearly to all questions I have addressed and improved the manuscript as expected

Reviewer #3 (Remarks to the Author):

The authors have responded adequately to all my comments. The manuscript is of very high quality and scientific relevance and I recommend it for publication as it stands.